# Automatic comprehensive radiological reports for clinical acute stroke MRIs

Chin-Fu Liu[1,2], Yi Zhao [3], Vivek Yedavalli[4], Richard Leigh [5], Vitor Falcao[6], on behalf of the STIR and VISTA Imaging investigators*, Michael I. Miller [1,2,7], Argye E. Hillis[5,8] & Andreia V. Faria [4✉]

**Abstract**

**Background** Although artificial intelligence systems that diagnosis among different conditions from medical images are long term aims, specific goals for automation of human-labor, time-consuming tasks are not only feasible but equally important. Acute conditions that require quantitative metrics, such as acute ischemic strokes, can greatly benefit by the consistency, objectiveness, and accessibility of automated radiological reports.

**Methods** We used 1,878 annotated brain MRIs to generate a fully automated system that outputs radiological reports in addition to the infarct volume, 3D digital infarct mask, and the feature vector of anatomical regions affected by the acute infarct. This system is associated to a deep-learning algorithm for segmentation of the ischemic core and to parcellation schemes defining arterial territories and classically-identified anatomical brain structures.

**Results** Here we show that the performance of our system to generate radiological reports was comparable to that of an expert evaluator. The weight of the components of the feature vectors that supported the prediction of the reports, as well as the prediction probabilities are outputted, making the pre-trained models behind our system interpretable. The system is publicly available, runs in real time, in local computers, with minimal computational requirements, and it is readily useful for non-expert users. It supports large-scale processing of new and legacy data, enabling clinical and translational research.

**Conclusion:** The generation of reports indicates that our fully automated system is able to extract quantitative, objective, structured, and personalized information from stroke MRIs.

**Plain language summary**

Artificial intelligence (AI) uses computer software to solve problems that normally require human input. It is likely that AI will take over, or help with, certain tasks in medical imaging, particularly where these tasks are time-consuming and laborious for clinicians. Here, we demonstrate the possibility of using AI to generate radiological reports for brain scans from patients who have had a stroke. These reports provide a summary of what is shown in the scans, and are normally written by clinicians. Our system performs similarly to human experts, is fast, publicly available, and runs on normal computers with minimal computational requirements, meaning that it might be a useful tool for researchers and clinicians to use when assessing and treating patients with stroke.

[1] Center for Imaging Science, Johns Hopkins University, Baltimore, MD, USA. [2] Department of Biomedical Engineering, Johns Hopkins University, Baltimore, MD, USA. [3] Department of Biostatistics and Health Data Science, Indiana University School of Medicine, Indianapolis, IN, USA. [4] Department of Radiology, School of Medicine, Johns Hopkins University, Baltimore, MD, USA. [5] Department of Neurology, School of Medicine, Johns Hopkins University, Baltimore, MD, USA. [6] Weiss Memorial Hospital, Chicago, IL, USA. [7] Kavli Neuroscience Discovery Institute, Johns Hopkins University, Baltimore, MD, USA. [8] Department of Physical Medicine & Rehabilitation, and Department of Cognitive Science, Johns Hopkins University, Baltimore, MD, USA. *A list of authors and their affiliations appears at the end of the paper. ✉email: afaria1@jhmi.edu

The advancement in labeling techniques signaled the end of services that require human interpretation of images, such as radiology reading. However, 6 years after the announcement of the "end of the path" for radiologists[1], they are still alive and operating. Humans still seem superior than machine to decode high level features and relate them to meaningful concepts. Radiologists might have some time until the massive annotated knowledge representing all the variation in human population and diseases will feed AI models that produce comprehensive results and could rival humans in all aspects.

The development of new unsupervised learning methods[2] or the massive labeling of medical images to train supervised methods are daunting projects. It is unlikely that multipurpose reporting systems, that can detect and differentiate among several conditions simultaneously, can be created at short term. However, specific goals for automation are not only feasible but also important[3]. For instance, the typical work flow for reporting quantitative data, e.g., performing a measure in Picture Archive and Communication System (PACS), is redundant, subjective, time-consuming and hard to record. Automated radiological reports describing consistent lesion features such as location, contrast, volumetric properties, and related effects could be a time-saver, particularly in acute conditions and in those that require quantitative report. In addition, it would produce text-structured information that would, in future, reduce the challenges of natural language processing (NLP) and other artificial intelligence (AI) applications in medical analysis.

The initial attempts to generate automated labels for medical images are based on AI models for automated recognition and classification of abnormalities[4,5]. The first systems to generate automated reports focused in specific goals and were trained in 2D images as chest X-rays[6] (please see[7] for a review) and mammography[8]. These are widely performed medical images, relatively less challenging for human annotation, compared to 3D MRIs. The possibility of aggregating considerable sized datasets of these images has been supporting the nascent development of deep learning models (DL) for report generation[9]. For 3D MRIs, the scarcity of large datasets and difficulties on expert annotation, as well as the unbalance between abnormal and normal cases to derive the knowledge about populations variation, impose extra challenges for AI. Finally, the current inability of AI models to provide findings as well as underlying justifications reduce their popularity among medical professionals.

We present an automated system, the Acute stroke detection and segmentation, ADS[10], to generate radiological reports for MRIs of patients with clinical diagnosis of acute ischemic stroke. This system was developed in a large database of annotated 1878 cases[11], associated to a deep-learning algorithm for detection and segmentation of the ischemic core in diffusion weighted images (DWIs)[12]. It reports the lesion location and volume in terms of arterial brain territories[13] and classical brain structures[14]. It can be combined to other brain segmentation schemes to generate reports in different sets of structures and scores of clinical importance, such as ASPECTS[15]. Most important, ADS is public, user-friendly, runs in CPU of local, regular personal computers with minimum computational requirements (as described previously[12] and in the tool documentation[10]), outputting the reports with a single command line. It therefore fulfills all the conditions to perform large scale, reliable and reproducible clinical and translational research.

## Methods

**Cohort and Images**. This study included MRIs of patients admitted to the Comprehensive Stroke Center at Johns Hopkins Hospital with the clinical diagnosis of ischemic stroke, between 2009 and 2019. This is a subset of the "Annotated Clinical MRIs and Linked Metadata of Patients with Acute Stroke", an anonymized dataset organized under waiver of patient consent (IRB00228775), publicly shared[11]. Briefly, the entire dataset consists of 2888 multimodal clinical MRIs performed at the admission of patients with acute brain strokes, retrospectively archived over 10 years, organized under FAIR principles[16]. Of note, only patients with MRI diagnosis of acute stroke were included, which represents a subset of all hospital stroke patients. The dataset includes lesion segmentation, expert radiological description, patient demographic information, and basic clinical profile. Details of this publicly available dataset are in the documentation that accompanies the data[11] and in the related publication[17]. We have complied with all relevant ethical regulations from the Johns Hopkins Institutional Review Board that approved this study (IRB00290649).

In this study, we included 1,878 mutually exclusive MRIs with evidence of ischemic stroke in the diffusion weighted images (DWI). The flowchart for data inclusion is shown in Fig. 1. The data were random split into training set ($n = 1414$, 75%) and testing set ($n = 464$, 25%). The detailed description of the demographic, lesion and scanner profiles of the data used in this study is in Table 1. The distribution of infarcts according to brain location and the demographic characteristics reflect the general population of stroke patients. MRIs were obtained on eleven scanners from four different manufacturers, in different magnetic fields (1.5 T and 3 T), with dozens of different protocols. The DWIs had high in plane (axial) resolution ($1.2 \times 1.2$ mm, or less), and typical clinical high slice thickness (up to 5 mm plus gap). Although a challenge for imaging processing, the technical heterogeneity promotes the potential generalization of the resulting developed tools.

Our testing set was completely independent and unseen in the machine learning training and validation phases. We reinforce that although we used data from a single National Stroke Center, these data originated from multiple hospitals and a large geographic region, reflecting the profile of the national population with stroke. Still, a second external testing set (STIR (http://stir.dellmed.utexas.edu/), $n = 100$), was used to test the generalization of our models in a unrelated population. We have complied with all relevant STIR regulations for data usage.

The delineation of the stroke core was defined in the DWI by 2 experienced evaluators and revised by a neuroradiologist until reaching a final decision by consensus, as described in[17]. The DWIs were mapped to a common template in MNI space[18] by 12-parameter linear deformation; the deformation matrix was then applied to the binary stroke masks. The detailed description of these procedures, including used parameters and quality control of the image mapping, is in our publication describing the dataset[17].

**Visual lesion description and validation**. The infarct location was classified by a neuroradiologist according to two schemes:

1. arterial territories, which consists of territories of the following arteries: anterior, medial, and posterior cerebral (ACA, MCA (excluding lenticulostriates)), and PCA (excluding choroidal and thalamoperforating), superior and inferior cerebellar, medial and lateral lenticulostriate, posterior and anterior choroidal and thalamoperforating, and the watershed zone ACA-MCA and MCA-PCA;
2. "classical" anatomy, which defines frontal, temporal, parietal, occipital lobes, insula, internal capsule, deep white matter (corona radiata and centrum semiovale), thalamus, basal ganglia, cerebellum, and brainstem (midbrain, pons, and medulla).

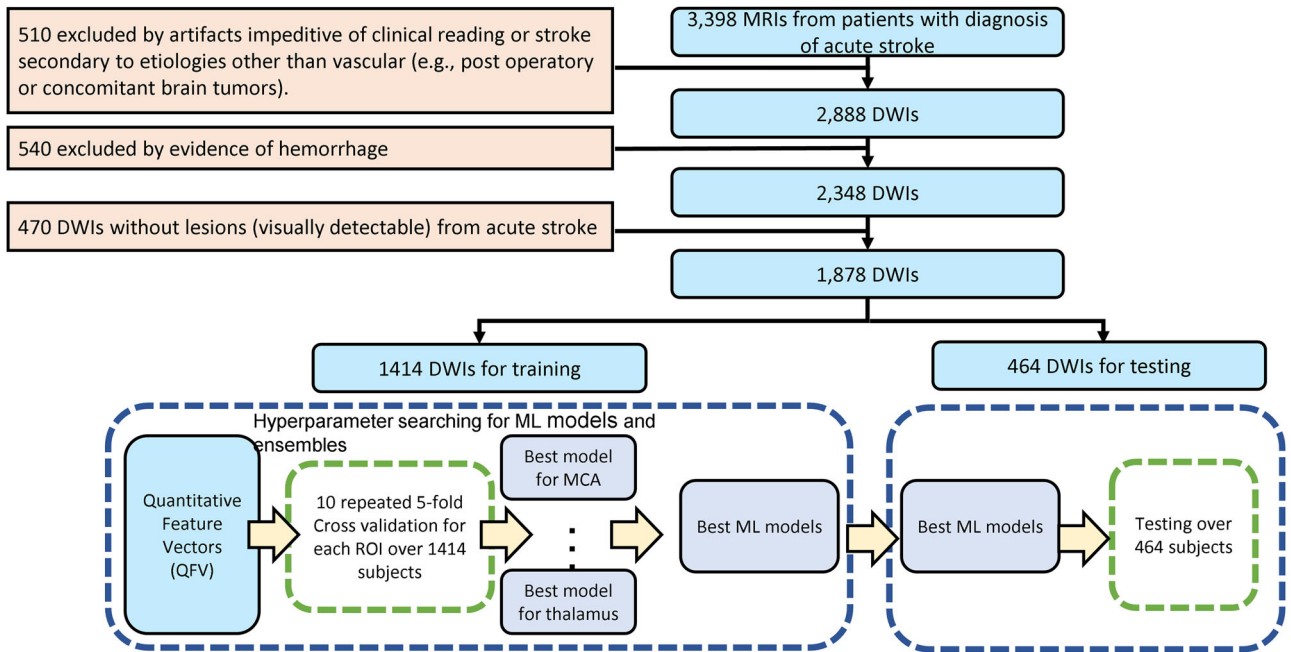

**Fig. 1 Data description and study design.** The flowchart describes data inclusion and exclusion, and of the design used for developing and testing of machine learning models.

Regions considered injured received a score of 1; the non-injured received 0. The evaluator also recorded the presence (1) or absence (0) of hydrocephalus. The lesion descriptions were validated against the clinical radiological reports in the medical records in a subsample of 110 of cases (6%). The "events" annotated, using BRAT rapid annotation tool (https://brat.nlplab.org/), consisted in words describing stroke type (e.g., "ischemic", "hemorrhage", "bleeding") and location (e.g., words related to arterial territories and brain structures). The annotations were as standardized as possible, to enable the comparison with our standardized description (e.g., "occipital lobe" became "occipital"; "middle cerebral artery" became "MCA"). The comparison between the annotation of the radiological reports and our descriptive metadata was made by the inter-annotator agreement (IAA) with Kohen's Kappa, using the "irr" R package (https://cran.r-project.org/web/packages/irr/index.html). Values for IAA Kappa range from 0 to 1 (1 is perfect agreement).

There was a total agreement for the description of stroke type (ischemic) between the clinical radiologcal reports and our metadata. For the lesion locations, the mean IAA Kappa was $0.71 \pm 0.16$, which is a high level of agreement. The indices varied from perfect agreement of 1 (for regions such as thalamus), to the lowest 0.5 (for parietal lobe). We note that the disagreements were, in their vast majority, result of semantic variations or analysis at different levels of granularity, rather than divergence in radiological evaluation. For instance, if the clinical radiological report says "perirolandic area" and our text-standardized description says "parietal" lobe, this was considered a disagreement, although the perirolandic area is part of the parietal lobe. Based on the results of IAA Kappa and these observations, we considered our radiological descriptions aligned with the medical records, and well suited for training the automated models.

**Multiple evaluators descriptions**. To access the level of variation in visual descriptions, and the agreement of different evaluators with the developed automated reports, two other clinical experts, a neuroradiologist (VY) and a neurologist (RL), with >10 years of experience in stroke care and image reading, classified the infarct location in the whole testing set ($n = 464$), following the same procedures described above. The comparison among the three evaluators, and among the evaluators and the automated classification was made by the intraclass correlation (ICC) using the function ICC3 of the Python package "pingouin.intraclass-corr"[19].

A second question is whether the automated radiological reports would aid the flow of clinical stroke care, particularly in settings that do not count on highly trained experts or second radiology readers full time in emergency service. Testing clinical impact is beyond the scope of this paper, as it depends on further stages of technical and bureaucratic technology development. Even so, as proof-of-concept, we asked one emergency room physician, not formally trained in neuro-radiology or neurology (VF), to classify the stroke location in a testing subset ($n = 155$), again using the same procedures described above. The results of the here called non-expert physician and the automated radiological reports were compared to the expert physicians' readings, and rated as "in agreement", "in partial agreement", or "in disagreement" with those. "In disagreement" was used if an infarcted area was not or was wrongly described, and that would have clinical implications, such as change of clinically relevant metrics (e.g. ASPECTS[20]). "In partial agreement" was used if the error would have no potential clinical implications. We also recorded the time for the non-expert physician reading.

**Automatic extraction of feature vectors**. The quantitative features used to train the models for automatic classification of the infarct location were defined to be compatible with the visual scoring. Digital atlases, based on arterial territories[13] (Arterial atlas—NITRC. https://www.nitrc.org/docman/?group_id=1498) and classical anatomy[14] (illustrated in Fig. 2), were overlaid on the brains in standardized space (MNI). These atlases define similar regions of interest (ROI) as those used in the visual analysis, which are the most clinically relevant for the description of the acute infarct location. The quantitative feature vectors

**Table 1 Population, lesion and scanner profiles. For continuous variables the numbers are shown as median [IQR stands for interquartile range].**

| Dataset | total | Training | Testing | *p*-value |
|---|---|---|---|---|
| **Number of Participants** | 1878 | 1414 | 464 | |
| **Age in years** | 62.0 [53,72] | 62.0 [52,72] | 62.0 [54,72] | 0.40 |
| **Sex** | | | | |
| male | 1012 (53.89%) | 756 (53.47%) | 256 (55.17%) | 0.56 |
| female | 866 (46.11%) | 658 (46.53%) | 208 (44.83%) | |
| **Race/Ethnicity** | | | | |
| African American | 824 (43.88%) | 601 (42.5%) | 223 (50.22%) | 0.79 |
| Caucasian | 533 (28.38%) | 393 (27.79%) | 140 (30.17%) | |
| Asian | 44 (2.34%) | 32 (2.26%) | 12 (2.59%) | |
| Missing Data | 477 (25.40%) | 388 (27.44%) | 89 (19.18%) | |
| **Lesioned hemisphere** | | | | |
| left | 834 (44.41%) | 635 (44.91%) | 199 (42.89%) | 0.09 |
| right | 766 (40.79%) | 554 (39.18%) | 212 (45.69%) | |
| bilateral | 278 (14.80%) | 225 (15.91%) | 53 (11.42%) | |
| **infarct location (arterial territory)** | | | | 0.012 |
| ACA | 98 (5.22%) | 73 (5.16%) | 25 (5.39%) | 0.945 |
| MCA | 969 (51.60%) | 709 (50.14%) | 260 (56.03%) | 0.032 |
| PCA | 257 (13.68%) | 193 (13.65%) | 64 (13.79%) | 1.000 |
| cerebellar | 255 (13.58%) | 196 (13.86%) | 59 (12.72%) | 0.584 |
| basilar | 113 (6.02%) | 95 (6.72%) | 18 (3.88%) | 0.034 |
| Lateral Lenticulostriates | 470 (25.03%) | 331 (23.41%) | 139 (29.96%) | 0.006 |
| Choroidal&Thalamoperforating | 313 (16.67%) | 243 (17.19%) | 70 (15.09%) | 0.327 |
| watershed | 209 (11.13%) | 170 (12.02%) | 39 (8.41%) | 0.039 |
| **infarct location (brain structure)** | | | | 0.014 |
| basal ganglia | 396 (21.09%) | 270 (19.09%) | 126 (27.16%) | 0.0004 |
| deep white matter | 716 (38.13%) | 539 (38.12%) | 177 (38.15%) | 0.979 |
| cerebellum | 260 (13.84%) | 201 (14.21%) | 59 (12.72%) | 0.435 |
| frontal lobe | 638 (33.97%) | 471 (33.31%) | 167 (35.99%) | 0.358 |
| insula | 323 (17.20%) | 239 (16.90%) | 84 (18.10%) | 0.638 |
| internal capsule | 184 (9.80%) | 145 (10.25%) | 39 (8.41%) | 0.267 |
| brainstem | 228 (12.14%) | 183 (12.94%) | 45 (9.70%) | 0.069 |
| occipital lobe | 287 (15.28%) | 222 (15.70%) | 65 (14.01%) | 0.393 |
| parietal lobe | 522 (27.80%) | 403 (28.50%) | 119 (25.65%) | 0.228 |
| temporal lobe | 423 (22.52%) | 312 (22.07%) | 111 (23.92%) | 0.481 |
| thalamus | 218 (11.61%) | 176 (12.45%) | 42 (9.05%) | 0.052 |
| **hydrocephalus** | 533 (28.38%) | 404 (28.57%) | 129 (27.80%) | 0.159 |
| **Lesion volume in ml** | 4.27 [0.98,21.98] | 4.18 [0.96,21.21] | 4.54 [1.04,27.67] | 0.066 |
| **MRI manufacturer** | | | | |
| Siemens | 1667 (88.76%) | 1229 (86.92%) | 438 (94.40%) | 0.0003 |
| Phillips | 15 (0.80%) | 13 (0.92%) | 2 (0.43%) | |
| GE | 166 (8.84%) | 144 (10.18%) | 22 (4.74%) | |
| others | 30 (1.60%) | 28 (1.98%) | 2 (0.43%) | |
| **MRI magnetic field** | | | | |
| 1.5 T | 1217 (64.88%) | 944 (66.76%) | 273 (58.84%) | 0.002 |
| 3.0 T | 661 (35.20%) | 470 (33.24%) | 191 (41.16%) | |
| **Voxel size in mm3** | | | | |
| Voxel size | 5.74 [2.52,7.60] | 5.70 [3.00,7.60] | 5.74 [2.33,7.40] | 0.43 |
| Height/Width | 1.20 [0.63,1.30] | 1.20 [0.90,1.38] | 1.20 [0.60,1.25] | |
| Thickness | 5.00 [4.0,5.0] | 5.00 [4.0,5.0] | 5.00 [4.0,5.0] | |

For discrete variables, the numbers are the counts and the percentage they represent from the total number in each class.

(QFV) extracted proportionally reflect the ratio of injury in each ROI (i.e., the proportion of ROI voxels in which the infarct mask = 1). We note that all ROIs are bilateral (except by the brainstem) and homologous ROIs have approximately the same volume. One illustrative example is shown in Fig. 3 and Table 2. The infarct volume (in log ml) was also included in QFVs.

We also trained and tested a model to predict hydrocephalus, as this is an important characteristic to be reported in strokes. Two strips of 5-voxel width bandwidth were defined around the five sub-regions of the lateral ventricles (LV, as defined in template brain[14]): the outside strip of the LV (OLV), and the inside strip of the LV (ILV). After linearly mapping the brain to the template, the number of voxels with ADC intensity >0.0018 mm$^2$/s (CSF voxels) and <0.0018 mm$^2$/s (non-CSF voxels)[21] are calculated to generate:

1 $\gamma_{OLVR}$: the ratio of the number of the deformed non-CSF voxels in OLV over the number of voxels in OLV. $\gamma_{OLVR}$ lower than the dataset average $\gamma_{OLVR}$ indicates ventricular enlargement compared to the expected average ventricle size (although not necessarily hydrocephalus).

2 $\gamma_{ILVR}$: the ratio of the number of the deformed CSF voxels in ILV over the number of voxels in ILV. $\gamma_{ILVR}$ lower than the dataset average $\gamma_{ILVR}$ indicates possible ventricular compression or midline shift

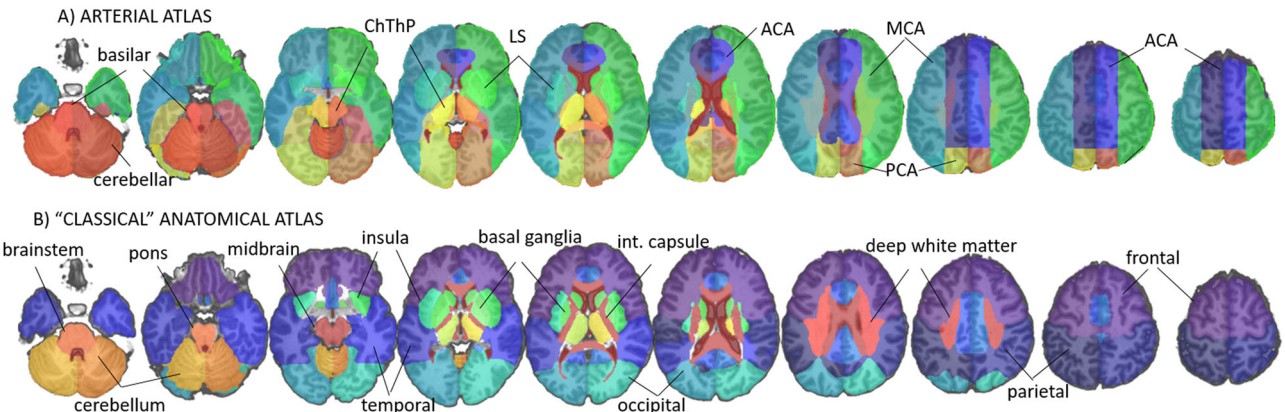

**Fig. 2 Atlases defining the arterial territories[13] (A) and brain structures[14] (B) used in this study.** The regions of interest (ROIs) are overlaid in the template[18] T1-WI.

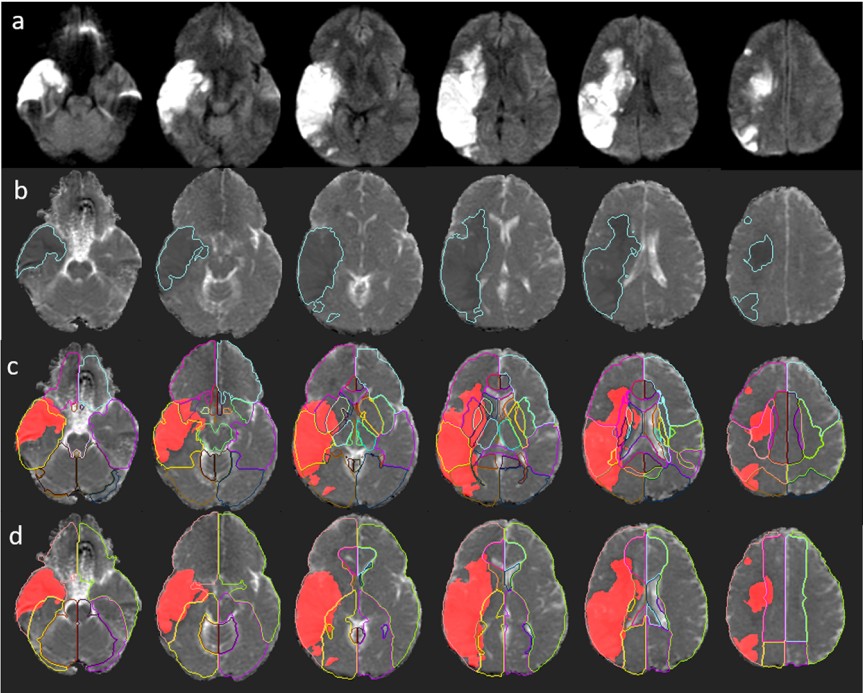

**Fig. 3 Illustrative example of the infarct location prediction to generate automated radiological report.** The figures show a large acute ischemic infarct in DWI (**a**). The infarct core, automatically segmented[12], is overlaid in ADC (**b**). Brain atlases representing classical anatomical structures (**c**) and arterial territories (**d**) allow to quantify the injury in diverse regions of interest (ROIs). The calculated quantitative feature vectors (QFV) are in Table 2.

To access the accuracy of the QFVs extracted, we deliver quality control indices (described in the supplementary material—QCI section) that indicate the agreement between the contour of the brain in question and the atlases in which the brain structures are defined. Lastly, we extracted QFVs from brains non-linearly mapped to the atlases (with Dipy), to evaluate the influence of the brain mapping method (linear vs. non-linear) in the automated prediction of stroke location.

**Machine Learning (ML) classification models to predict infarct location**. We developed, validated, and tested seven models (described below) to predict the infarct location using the QFV calculated with the human-segmented lesions. All ML models were cross validated over the training set (1414 individuals, 75%) for hyperparameter searching and tested in the completely independent testing set of 464 individuals (flowchart in Fig. 1).

We performed 5-fold cross validation on the training set, for a large set of searching parameters. The models' hyperparameters with the top 3 performances (according to the sum of balanced accuracy (BACC) and F1 score, from this first run of 5-fold cross validation) were further determined and selected via 10 repeat 5-fold cross-validation, on the training set. The details of searching parameters' sets, final optimal parameters, cross validation results, and the definitions of performance metrics are in the Supplementary Methods and Supplementary Data 1 and 2.

The simplest model, the Binary Threshold (BT), was built to classify the stroke location via thresholding its corresponding ROI component in the QFV for each participant. The threshold can be interpreted as the minimum percentage of the ROI occupied by the infarct mask to lead its classification as injured (and receive a score of 1). The threshold for each ROI was the minimal level to achieve the highest sum of the BACC and F1 score, found by cross-validation of the training set. The optimal thresholds for

**Table 2 Calculation of quantitative feature vectors (QFV) shown in Fig. 3.**

| | basal ganglia | deep wm | cerebellum | frontal lobe | insula | int. capsule | brainstem | occipital lobe | parietal lobe | temporal lobe | thalamus |
|---|---|---|---|---|---|---|---|---|---|---|---|
| visual | 1 | 1 | 0 | 1 | 1 | 0 | 0 | 0 | 1 | 1 | 0 |
| infarct volume | 7262 | 15693 | 0 | 31604 | 14917 | 6967 | 818 | 22219 | 27963 | 96580 | 1695 |
| ROI volume | 11959 | 19655 | 103009 | 249674 | 16323 | 8061 | 44211 | 100804 | 137367 | 132439 | 10077 |
| QFV | 0.61 | 0.79 | 0 | 0.13 | 0.91 | 0.86 | 0.02 | 0.22 | 0.2 | 0.73 | 0.17 |
| predict. prob. | 0.72 | 0.79 | 0.01 | 0.74 | 0.81 | 0.04 | 0.14 | 0.16 | 0.86 | 0.77 | 0.09 |
| prediction | 1 | 1 | 0 | 1 | 1 | 0 | 0 | 0 | 1 | 1 | 0 |

| | ACA | MCA | PCA | cerebellar | basilar | Lat. Lenticul. | Chor&Thal.Perf |
|---|---|---|---|---|---|---|---|
| visual | 0 | 1 | 0 | 0 | 0 | 1 | 0 |
| infarct volume | 10294 | 199607 | 6856 | 0 | 12 | 20540 | 6661 |
| ROI volume | 202742 | 437010 | 119009 | 108869 | 18168 | 26842 | 28648 |
| QFV | 0.05 | 0.46 | 0.06 | 0 | 0 | 0.77 | 0.23 |
| predict. prob. | 0.01 | 0.98 | 0.05 | 0.01 | 0 | 0.54 | 0.05 |
| prediction | 0 | 1 | 0 | 0 | 0 | 1 | 0 |

Each QFV component represents the proportion of the respective ROI affected by the infarct. The rows visual show the classification of the respective ROI in injured or not (1 or 0) according to expert human evaluation, which is here considered the gold standard. The prediction rows show the infarct location predicted by ML, based on the QFVs. At the bottom, the comparison between the real radiological report in medical records and the predicted report. These are all outputs of our automated tool, ADS[10]
Radiological report from medical records: Large area of restricted diffusion in the right middle cerebral artery territory mainly involving the right temporal lobe, right parietal lobe, and portions of the right frontal lobe. The area measures 11 cm by 4.8 cm. The size of the ventricles is compatible with the age.
Predicted radiological report: Area of restricted diffusion within the right brain hemisphere, with 165.195 ml, in the territory of the middle cerebral artery and possibly Lateral Lenticulostriate. The area involves the following brain regions: basal ganglia, deep white matter, frontal lobe, insula, parietal lobe, and temporal lobe. There is no hydrocephalus. The predicted MCA-ASPECTS is 2.

each ROI are summarized in the Supplementary Data 1. The remaining six models were Linear Discriminant Analysis (LDA), Quadratic Discriminant Analysis (QDA), Random Forest (RF), K-nearest Neighbors (KNN), Support Vector Machine (SVM), and Multi-layer Perceptron (MLP). The measures of models' performances are described in the Table 3.

Because some ROIs are injured only in a few cases (see Table 1) over our large dataset, deep neural network models would suffer from imbalanced classes issue. In addition, the interpretation of deep learning models is not as straightforward as that of classic ML methods. Furthermore, given that the ML models employed proved sufficiently efficient compared to humans (as we will show in Results), we opted to further investigate the potential of deep learning techniques to improve the classification performance when this dataset is expanded, or when other public sets of clinical data become available.

**Feature analysis**. The implementation of interpretable models potentially increases their practical usefulness and enable to investigate whether machine uses features of biological relevance, similarly to humans. We explored how the visual classification attributed to different ROIs relate to each other, as well as the relationship between visual analysis and QFV for each ROI, and between different QFV components, using bivariate correlations (Table 4).

We then conducted an analysis of feature importance to inspect how the ML models use the QFVs to describe lesion location. The analyses presented here are based on Random Forest (RF) models, which had the best average performance (BACC, F1) among all ML models. The impurity-based feature importance analysis[22,23] was conducted to build 100 RF models on the training set with the optimal parameters selected from cross validation. The Mean Decrease in Impurity (MDI), shown in Fig. 4, indicates the feature importance (high MDIs correspond to the most important features). We also conducted permutation feature importance tests (100 iterations), using the training and testing set separately (Supplementary Tables 1 and 2 and Supplementary Fig. 1 and 2), to illustrate the consistency in feature learning and their potential generalization, respectively.

To generate and deliver explanations about the predictions in a new given sample, we adapted the SHapley Additive exPlanations (SHAP)[24] module in the ADS. The SHAP computes the Shapley values[25] of features via coalitional game theory to indicate how to fairly distribute prediction of an instance among features. Because the Shapley feature value is linearly additive, this value can be directly added or subtracted from the probability of predicts, making the models' interpretation straightforward. The outputs are intuitively comprehensible graphs of the predicted infarct location (example in Supplementary Fig. 3), explaining how the QFV components were combined to predict injury in each brain area.

**Reporting summary**. Further information on research design is available in the Nature Portfolio Reporting Summary linked to this article.

## Results
**Correlations among human expert evaluation and QFVs**. Table 4 illustrates how the visual determination of the presence of infarct (yes = 1, no = 0) is related in different ROIs. High correlation between a pair ROIs means that the infarct tends to co-exist on them. As expected, given the spatial coalescence of infarct lesions, neighboring ROIs defined by the classical anatomic atlas (Table 4.II.a) are the highest correlated (e.g., frontal, parietal, and temporal; which are part of the MCA territory). Compared to those, ROIs defined by the arterial territory atlas (Table 4.I.a) are less correlated, since they follow the distribution of the infarcts by definition. Table 4.I.b and II.b demonstrate the correlation between QFV components. They represent the quantitative version of the qualitative scores in Table 4.I.a and II.a, to which they highly agree. This indicates that the quantitative information coded in the QFVs (the proportion of each ROI affected by the infarct) is likely reflecting the qualitative information that humans relay in their visual analysis. Similar to the visual analysis, each QFV component is highly correlated to the QFV components of its neighbor ROIs.

Table 4.I.c and II.c combine the information above, showing the correlation between visual analysis and QFV components.

**Table 3 Comparison of models performances to predict infarct location in the testing set (*n* = 464). The numbers in parenthesis are frequency of infarcts per region.**

| | LDA | QDA | KNN | SVM | RF | MLP | BT |
|---|---|---|---|---|---|---|---|
| **I. According to arterial territory** | | | | | | | |
| **ACA (25)** | | | | | | | |
| BACC | 0.638 | 0.696 | 0.659 | 0.672 | 0.712 | 0.746 | 0.823 |
| F1 | 0.412 | 0.213 | 0.346 | 0.439 | 0.512 | 0.52 | 0.404 |
| Precision | 0.778 | 0.128 | 0.333 | 0.563 | 0.611 | 0.52 | 0.275 |
| Sensitivity | 0.28 | 0.64 | 0.36 | 0.360 | 0.44 | 0.52 | 0.76 |
| Kappa | 0.394 | 0.136 | 0.307 | 0.414 | 0.489 | 0.493 | 0.353 |
| **MCA (260)** | | | | | | | |
| BACC | 0.773 | 0.785 | 0.85 | 0.887 | 0.913 | 0.883 | 0.815 |
| F1 | 0.746 | 0.748 | 0.866 | 0.888 | 0.916 | 0.88 | 0.84 |
| Precision | 0.897 | 0.936 | 0.875 | 0.940 | 0.954 | 0.951 | 0.833 |
| Sensitivity | 0.638 | 0.623 | 0.858 | 0.842 | 0.881 | 0.819 | 0.846 |
| Kappa | 0.525 | 0.544 | 0.699 | 0.763 | 0.818 | 0.751 | 0.632 |
| **PCA (64)** | | | | | | | |
| BACC | 0.707 | 0.778 | 0.753 | 0.822 | 0.87 | 0.831 | 0.837 |
| F1 | 0.574 | 0.673 | 0.653 | 0.757 | 0.828 | 0.775 | 0.637 |
| Precision | 0.9 | 0.804 | 0.892 | 0.894 | 0.923 | 0.915 | 0.538 |
| Sensitivity | 0.422 | 0.578 | 0.516 | 0.656 | 0.75 | 0.672 | 0.781 |
| Kappa | 0.533 | 0.63 | 0.615 | 0.725 | 0.803 | 0.745 | 0.566 |
| **cerebellar (59)** | | | | | | | |
| BACC | 0.626 | 0.785 | 0.725 | 0.823 | 0.948 | 0.81 | 0.942 |
| F1 | 0.4 | 0.649 | 0.577 | 0.750 | 0.893 | 0.747 | 0.794 |
| Precision | 0.938 | 0.692 | 0.737 | 0.867 | 0.871 | 0.925 | 0.683 |
| Sensitivity | 0.254 | 0.61 | 0.475 | 0.661 | 0.915 | 0.627 | 0.949 |
| Kappa | 0.366 | 0.601 | 0.531 | 0.719 | 0.876 | 0.719 | 0.759 |
| **basilar (18)** | | | | | | | |
| BACC | 0.609 | 0.777 | 0.681 | 0.817 | 0.844 | 0.763 | 0.952 |
| F1 | 0.333 | 0.175 | 0.378 | 0.533 | 0.565 | 0.488 | 0.456 |
| Precision | 0.667 | 0.097 | 0.368 | 0.444 | 0.464 | 0.435 | 0.295 |
| Sensitivity | 0.222 | 0.889 | 0.389 | 0.667 | 0.722 | 0.556 | 1 |
| Kappa | 0.32 | 0.113 | 0.353 | 0.511 | 0.544 | 0.464 | 0.421 |
| **Lenticulostriate (139)** | | | | | | | |
| BACC | 0.712 | 0.693 | 0.833 | 0.854 | 0.859 | 0.865 | 0.854 |
| F1 | 0.595 | 0.581 | 0.784 | 0.814 | 0.812 | 0.824 | 0.757 |
| Precision | 0.924 | 0.427 | 0.883 | 0.904 | 0.85 | 0.878 | 0.633 |
| Sensitivity | 0.439 | 0.906 | 0.705 | 0.741 | 0.777 | 0.777 | 0.942 |
| Kappa | 0.498 | 0.293 | 0.706 | 0.746 | 0.737 | 0.756 | 0.622 |
| **Choroidal & Thalamoperfurating (70)** | | | | | | | |
| BACC | 0.578 | 0.679 | 0.751 | 0.734 | 0.802 | 0.784 | 0.72 |
| F1 | 0.273 | 0.479 | 0.584 | 0.581 | 0.672 | 0.638 | 0.434 |
| Precision | 0.667 | 0.569 | 0.597 | 0.667 | 0.687 | 0.647 | 0.309 |
| Sensitivity | 0.171 | 0.414 | 0.571 | 0.514 | 0.657 | 0.629 | 0.729 |
| Kappa | 0.225 | 0.403 | 0.512 | 0.517 | 0.615 | 0.574 | 0.282 |
| **hydrocephalus (129)** | | | | | | | |
| BACC | 0.872 | 0.827 | 0.824 | 0.849 | 0.840 | 0.842 | |
| F1 | 0.819 | 0.713 | 0.762 | 0.790 | 0.787 | 0.784 | |
| Precision | 0.832 | 0.594 | 0.827 | 0.824 | 0.855 | 0.828 | NA |
| Sensitivity | 0.806 | 0.884 | 0.705 | 0.760 | 0.729 | 0.744 | |
| Kappa | 0.751 | 0.570 | 0.679 | 0.714 | 0.713 | 0.706 | |
| **II. According to classical brain structures** | | | | | | | |
| **basal ganglia (126)** | | | | | | | |
| BACC | 0.75 | 0.797 | 0.838 | 0.830 | 0.868 | 0.857 | 0.874 |
| F1 | 0.663 | 0.711 | 0.798 | 0.775 | 0.818 | 0.807 | 0.782 |
| Precision | 0.955 | 0.741 | 0.946 | 0.871 | 0.853 | 0.857 | 0.693 |
| Sensitivity | 0.508 | 0.683 | 0.69 | 0.698 | 0.786 | 0.762 | 0.897 |
| Kappa | 0.585 | 0.609 | 0.738 | 0.704 | 0.754 | 0.74 | 0.686 |
| **deep WM (177)** | | | | | | | |
| BACC | 0.668 | 0.687 | 0.71 | 0.763 | 0.811 | 0.774 | 0.773 |
| F1 | 0.542 | 0.657 | 0.637 | 0.702 | 0.766 | 0.718 | 0.728 |
| Precision | 0.75 | 0.508 | 0.661 | 0.779 | 0.716 | 0.756 | 0.604 |
| Sensitivity | 0.424 | 0.932 | 0.616 | 0.638 | 0.825 | 0.684 | 0.915 |
| Kappa | 0.367 | 0.323 | 0.426 | 0.546 | 0.605 | 0.558 | 0.497 |

**Table 3 (continued)**

| | LDA | QDA | KNN | SVM | RF | MLP | BT |
|---|---|---|---|---|---|---|---|
| **cerebellum (59)** | | | | | | | |
| BACC | 0.619 | 0.789 | 0.75 | 0.806 | 0.922 | 0.821 | 0.941 |
| F1 | 0.384 | 0.638 | 0.62 | 0.725 | 0.864 | 0.776 | 0.789 |
| Precision | 1 | 0.649 | 0.756 | 0.860 | 0.864 | 0.974 | 0.675 |
| Sensitivity | 0.237 | 0.627 | 0.525 | 0.627 | 0.864 | 0.644 | 0.949 |
| Kappa | 0.352 | 0.586 | 0.576 | 0.693 | 0.845 | 0.75 | 0.752 |
| **frontal (167)** | | | | | | | |
| BACC | 0.699 | 0.72 | 0.807 | 0.812 | 0.874 | 0.814 | 0.849 |
| F1 | 0.578 | 0.631 | 0.752 | 0.762 | 0.835 | 0.764 | 0.796 |
| Precision | 0.878 | 0.727 | 0.733 | 0.836 | 0.809 | 0.816 | 0.707 |
| Sensitivity | 0.431 | 0.557 | 0.772 | 0.701 | 0.862 | 0.719 | 0.91 |
| Kappa | 0.447 | 0.463 | 0.607 | 0.646 | 0.737 | 0.645 | 0.657 |
| **insula (84)** | | | | | | | |
| BACC | 0.82 | 0.867 | 0.853 | 0.860 | 0.901 | 0.89 | 0.912 |
| F1 | 0.72 | 0.721 | 0.743 | 0.754 | 0.787 | 0.789 | 0.763 |
| Precision | 0.753 | 0.628 | 0.714 | 0.725 | 0.712 | 0.74 | 0.642 |
| Sensitivity | 0.69 | 0.845 | 0.774 | 0.786 | 0.881 | 0.845 | 0.94 |
| Kappa | 0.662 | 0.648 | 0.683 | 0.697 | 0.734 | 0.738 | 0.698 |
| **internal capsule (39)** | | | | | | | |
| BACC | 0.613 | 0.679 | 0.743 | 0.780 | 0.752 | 0.757 | 0.768 |
| F1 | 0.297 | 0.234 | 0.512 | 0.613 | 0.56 | 0.537 | 0.324 |
| Precision | 0.314 | 0.135 | 0.488 | 0.639 | 0.583 | 0.512 | 0.2 |
| Sensitivity | 0.282 | 0.872 | 0.538 | 0.590 | 0.538 | 0.564 | 0.846 |
| Kappa | 0.237 | 0.103 | 0.465 | 0.579 | 0.521 | 0.492 | 0.217 |
| **brainstem (45)** | | | | | | | |
| BACC | 0.677 | 0.802 | 0.858 | 0.924 | 0.942 | 0.904 | 0.905 |
| F1 | 0.516 | 0.366 | 0.776 | 0.848 | 0.845 | 0.841 | 0.678 |
| Precision | 0.941 | 0.226 | 0.825 | 0.830 | 0.788 | 0.86 | 0.548 |
| Sensitivity | 0.356 | 0.956 | 0.733 | 0.867 | 0.911 | 0.822 | 0.889 |
| Kappa | 0.489 | 0.248 | 0.754 | 0.831 | 0.827 | 0.824 | 0.634 |
| **occipital (65)** | | | | | | | |
| BACC | 0.682 | 0.709 | 0.749 | 0.734 | 0.794 | 0.79 | 0.776 |
| F1 | 0.495 | 0.493 | 0.624 | 0.562 | 0.69 | 0.672 | 0.513 |
| Precision | 0.65 | 0.478 | 0.773 | 0.607 | 0.784 | 0.741 | 0.393 |
| Sensitivity | 0.4 | 0.508 | 0.523 | 0.523 | 0.615 | 0.615 | 0.738 |
| Kappa | 0.435 | 0.407 | 0.576 | 0.497 | 0.646 | 0.625 | 0.405 |
| **parietal (119)** | | | | | | | |
| BACC | 0.667 | 0.708 | 0.714 | 0.721 | 0.831 | 0.768 | 0.812 |
| F1 | 0.503 | 0.567 | 0.576 | 0.594 | 0.735 | 0.658 | 0.691 |
| Precision | 0.719 | 0.579 | 0.6 | 0.677 | 0.694 | 0.67 | 0.604 |
| Sensitivity | 0.387 | 0.555 | 0.555 | 0.529 | 0.782 | 0.647 | 0.807 |
| Kappa | 0.394 | 0.421 | 0.438 | 0.477 | 0.636 | 0.543 | 0.562 |
| **temporal (111)** | | | | | | | |
| BACC | 0.775 | 0.78 | 0.769 | 0.798 | 0.848 | 0.831 | 0.849 |
| F1 | 0.684 | 0.664 | 0.67 | 0.709 | 0.76 | 0.75 | 0.701 |
| Precision | 0.805 | 0.661 | 0.767 | 0.768 | 0.737 | 0.771 | 0.567 |
| Sensitivity | 0.595 | 0.667 | 0.595 | 0.658 | 0.784 | 0.73 | 0.919 |
| Kappa | 0.603 | 0.557 | 0.583 | 0.626 | 0.681 | 0.674 | 0.575 |
| **thalamus (42)** | | | | | | | |
| BACC | 0.637 | 0.776 | 0.82 | 0.873 | 0.867 | 0.883 | 0.859 |
| F1 | 0.407 | 0.542 | 0.691 | 0.790 | 0.744 | 0.795 | 0.526 |
| Precision | 0.706 | 0.481 | 0.718 | 0.821 | 0.727 | 0.805 | 0.379 |
| Sensitivity | 0.286 | 0.619 | 0.667 | 0.762 | 0.762 | 0.786 | 0.857 |
| Kappa | 0.374 | 0.49 | 0.662 | 0.770 | 0.718 | 0.775 | 0.457 |

They indicate more directly how humans inconspicuously use the quantitative information about the spatial distribution of the infarct (reflected by the QFV) to determine the infarct injury. The rows indicate that the visual classification attributed to each ROI is mostly correlated to the QFV component that corresponds to the ROI in question, and secondly, to the QFV components corresponding to neighboring ROIs, as expected. Again, as the infarct lesions extend beyond the artificial limits of the semantically defined areas, the human evaluation is not purely

**Table 4 Correlations among human classification of infarct location and quantitative feature vectors (QFVs) automatically extracted (*n* = 1414).**

**I According to Arterial Territory**

| | ACA | MCA | PCA | cerebellar | basilar | Lat Lent | Ch&ThPerf |
|---|---|---|---|---|---|---|---|
| **(a) visual vs. visual** | | | | | | | |
| ACA | 1 | −0.055 | −0.009 | −0.029 | −0.037 | −0.046 | −0.064 |
| MCA | −0.055 | 1 | −0.213 | −0.214 | −0.224 | −0.144 | −0.371 |
| PCA | −0.009 | −0.213 | 1 | 0.192 | −0.057 | −0.152 | 0.141 |
| cerebellar | −0.029 | −0.214 | 0.192 | 1 | −0.018 | −0.144 | 0.056 |
| basilar | −0.037 | −0.224 | −0.057 | −0.018 | 1 | −0.115 | −0.062 |
| Lat Lent | −0.046 | −0.144 | −0.152 | −0.144 | −0.115 | 1 | −0.194 |
| Ch&ThPerf | −0.064 | −0.371 | 0.141 | 0.056 | −0.062 | −0.194 | 1 |
| **(b) QFV vs. QFV** | | | | | | | |
| ACA | 1 | 0.538 | 0.138 | −0.026 | −0.03 | 0.333 | 0.129 |
| MCA | 0.538 | 1 | 0.16 | −0.049 | −0.064 | 0.706 | 0.238 |
| PCA | 0.138 | 0.16 | 1 | 0.207 | 0.149 | 0.059 | 0.661 |
| cerebellar | −0.026 | −0.049 | 0.207 | 1 | 0.716 | −0.061 | 0.288 |
| basilar | −0.03 | −0.064 | 0.149 | 0.716 | 1 | −0.068 | 0.301 |
| Lat Lent | 0.333 | 0.706 | 0.059 | −0.061 | −0.068 | 1 | 0.236 |
| Ch&ThPerf | 0.129 | 0.238 | 0.661 | 0.288 | 0.301 | 0.236 | 1 |
| **(c) QFV vs. visual** | | | | | | | |
| ACA | 0.312 | 0.042 | −0.016 | −0.043 | −0.026 | 0.025 | −0.041 |
| MCA | 0.15 | 0.377 | −0.091 | −0.158 | −0.175 | 0.209 | −0.1 |
| PCA | −0.001 | −0.091 | 0.52 | 0.17 | 0.133 | −0.105 | 0.318 |
| cerebellar | −0.035 | −0.09 | 0.086 | 0.476 | 0.318 | −0.123 | 0.069 |
| basilar | −0.057 | −0.103 | −0.037 | −0.008 | 0.207 | −0.098 | −0.005 |
| Lat Lent | −0.013 | 0.066 | −0.107 | −0.092 | −0.091 | 0.43 | −0.005 |
| Ch&ThPerf | −0.067 | −0.147 | 0.138 | 0.056 | 0.184 | −0.132 | 0.299 |

**II According to major brain structures**

| | basal ganglia | deep WM | cerebellum | frontal | insula | int capsule | brainstem | occipital | parietal lobe | temporal | thalamus |
|---|---|---|---|---|---|---|---|---|---|---|---|
| **(a) visual vs. visual** | | | | | | | | | | | |
| basal ganglia | 1 | 0.36 | −0.126 | 0.027 | 0.151 | 0.144 | −0.144 | −0.121 | −0.008 | 0.089 | −0.123 |
| deep WM | 0.36 | 1 | −0.203 | 0.085 | 0.128 | 0.104 | −0.251 | −0.127 | 0.075 | 0.039 | −0.19 |
| cerebellum | −0.126 | −0.203 | 1 | −0.124 | −0.108 | −0.091 | 0.066 | 0.153 | −0.078 | −0.104 | 0.031 |
| frontal | 0.027 | 0.085 | −0.124 | 1 | 0.354 | −0.17 | −0.232 | −0.037 | 0.278 | 0.311 | −0.226 |
| insula | 0.151 | 0.128 | −0.108 | 0.354 | 1 | −0.097 | −0.174 | −0.075 | 0.255 | 0.475 | −0.142 |
| int capsule | 0.144 | 0.104 | −0.091 | −0.17 | −0.097 | 1 | −0.123 | −0.095 | −0.126 | −0.107 | −0.0003 |
| brainstem | −0.144 | −0.251 | 0.066 | −0.232 | −0.174 | −0.123 | 1 | −0.068 | −0.211 | −0.149 | 0.046 |
| occipital | −0.121 | −0.127 | 0.153 | −0.037 | −0.075 | −0.095 | −0.068 | 1 | 0.132 | 0.028 | 0.167 |
| parietal | −0.008 | 0.075 | −0.078 | 0.278 | 0.255 | −0.126 | −0.211 | 0.132 | 1 | 0.344 | −0.148 |
| temporal | 0.089 | 0.039 | −0.104 | 0.311 | 0.475 | −0.107 | −0.149 | 0.028 | 0.344 | 1 | −0.103 |
| thalamus | −0.123 | −0.19 | 0.031 | −0.226 | −0.142 | −0.0003 | 0.046 | 0.167 | −0.148 | −0.102 | 1 |
| **(b) QFV vs. QFV** | | | | | | | | | | | |
| basal ganglia | 1 | 0.648 | −0.049 | 0.608 | 0.714 | 0.907 | −0.029 | 0.163 | 0.453 | 0.526 | 0.317 |
| deep WM | 0.648 | 1 | −0.067 | 0.742 | 0.724 | 0.723 | −0.055 | 0.263 | 0.773 | 0.648 | 0.243 |
| cerebellum | −0.049 | −0.067 | 1 | −0.045 | −0.068 | −0.051 | 0.647 | 0.144 | −0.046 | 0.007 | 0.126 |
| frontal | 0.608 | 0.742 | −0.045 | 1 | 0.727 | 0.599 | −0.04 | 0.214 | 0.733 | 0.59 | 0.225 |
| insula | 0.714 | 0.724 | −0.068 | 0.727 | 1 | 0.721 | −0.059 | 0.179 | 0.611 | 0.697 | 0.188 |
| int capsule | 0.907 | 0.723 | −0.051 | 0.599 | 0.721 | 1 | −0.017 | 0.216 | 0.542 | 0.611 | 0.415 |
| brainstem | −0.029 | −0.055 | 0.647 | −0.04 | −0.059 | −0.017 | 1 | 0.165 | −0.037 | 0.04 | 0.233 |
| occipital | 0.163 | 0.263 | 0.144 | 0.214 | 0.179 | 0.216 | 0.165 | 1 | 0.407 | 0.492 | 0.395 |
| parietal | 0.453 | 0.773 | −0.046 | 0.733 | 0.611 | 0.542 | −0.037 | 0.407 | 1 | 0.722 | 0.2 |
| temporal | 0.526 | 0.648 | 0.007 | 0.59 | 0.697 | 0.611 | 0.04 | 0.492 | 0.722 | 1 | 0.326 |
| thalamus | 0.317 | 0.243 | 0.126 | 0.225 | 0.188 | 0.415 | 0.233 | 0.395 | 0.2 | 0.326 | 1 |
| **(c) QFV vs. visual** | | | | | | | | | | | |
| basal ganglia | 0.548 | 0.305 | −0.087 | 0.215 | 0.331 | 0.532 | −0.07 | −0.015 | 0.119 | 0.176 | 0.088 |
| deep WM | 0.214 | 0.272 | −0.15 | 0.125 | 0.169 | 0.259 | −0.133 | −0.07 | 0.064 | 0.037 | −0.026 |
| cerebellum | −0.103 | −0.123 | 0.469 | −0.081 | −0.117 | −0.116 | 0.285 | 0.051 | −0.075 | −0.079 | 0.037 |
| frontal | 0.194 | 0.324 | −0.107 | 0.409 | 0.383 | 0.155 | −0.113 | 0.039 | 0.305 | 0.242 | −0.038 |
| insula | 0.348 | 0.462 | −0.077 | 0.421 | 0.683 | 0.362 | −0.081 | 0.07 | 0.41 | 0.417 | 0.039 |
| int capsule | 0.08 | −0.013 | −0.06 | −0.041 | −0.037 | 0.187 | −0.059 | −0.056 | −0.047 | −0.038 | 0.043 |
| brainstem | −0.119 | −0.162 | 0.072 | −0.125 | −0.156 | −0.138 | 0.411 | −0.066 | −0.125 | −0.105 | 0.033 |
| occipital | −0.036 | −0.011 | 0.095 | 0.012 | −0.044 | −0.046 | 0.092 | 0.509 | 0.079 | 0.096 | 0.209 |
| parietal | 0.158 | 0.311 | −0.074 | 0.277 | 0.276 | 0.171 | −0.101 | 0.223 | 0.453 | 0.363 | 0.031 |

**Table 4 (continued)**

**II According to major brain structures**

| | basal ganglia | deep WM | cerebellum | frontal | insula | int capsule | brainstem | occipital | parietal lobe | temporal | thalamus |
|---|---|---|---|---|---|---|---|---|---|---|---|
| temporal | 0.314 | 0.403 | −0.057 | 0.398 | 0.521 | 0.328 | −0.057 | 0.175 | 0.431 | 0.501 | 0.097 |
| thalamus | −0.069 | −0.122 | 0.078 | −0.091 | −0.115 | −0.036 | 0.158 | 0.132 | −0.1 | −0.027 | 0.457 |

Correlation matrices of the visual classification attributed to different ROIs (injured or not) (a), between the quantitative information in the QFVs (b), and between the visual classification and the QFVs (c). The regions in question follow either the distribution of arterial territories (I) or classic brain structures (II). Note that for each ROI, the highest correlation between visual classification and QFVs (c) is in general found in its corresponding QFV component, followed by QFV components of the adjacent ROIs.

based on how individual areas are affected, but also in the regional lesion pattern.

**Accuracy of ML models to predict infarct location**. The performance of models to predict the stroke location in the testing set is summarized in Table 3. The best models, in BACC and F1, were those created with random forest (RF), achieving an excellent agreement with the visual analysis (vast majority of BACC > 0.8). The lowest agreement, while still satisfactory, occurred in the ACA (considering the arterial territory scheme), and the internal capsule (considering the classical anatomical scheme). The most efficient RF model, retrained with the automated infarct segmentations[12], was included in our deployed pipeline to generate reports in ADS[10]. The hyperparameters and performances on cross-validation in the training set and in the testing set, using ADS infarct segmentation, are reported in Supplementary Data 3, 4 and 5.

As the infarct volume and location are correlated (e.g., small defined areas, such as the thalamoperfurating territory, irrigated by arteries of small caliber, tend to have small strokes), large ROIs had, in general, slightly better accuracy performance for all ML models. There was no significant difference in the prediction accuracy regarding the patient sex (male or female) or race (Black/ African America or Caucasian), time from stroke onset (> or <6 h), magnetic field (1.5 T or 3 T), and infarct side (left or right).

The performance of the RF model was slightly higher when using non-linearly normalized brain images (as shown in Supplementary Table 3), compared to linearly normalized. The model was robust in the external unrelated population (from STIR), demonstrating similar performance to that achieved in our independent testing set (results shown in Supplementary Table 4). The automated classification of infarct location was also robust when compared with that of different experts. The mean ICCs of the model against each of the three evaluators were $0.82 \pm 0.08$, $0.77 \pm 0.11$, and $0.81 \pm 0.08$, which rivaled to the ICCs among pairs of inter-evaluators: $0.75 \pm 0.12$, $0.8 \pm 0.09$, and $0.81 \pm 0.08$; with standard deviations consistently lower. The indices of agreement are presented in details and categorized by location in Supplementary Table 5. The regional distribution of ICCs was consistent inter-evaluators and between the model and the evaluators, i.e., regions with the lowest concordance among the model and the evaluators (e.g., ACA and internal capsule) also had the lowest concordance inter-evaluators (Supplementary Table 5 and Fig. 5). Of note, these regions correspond to those with lowest lesion frequency or those with unclear or less consensual boundaries.

Our system was more accurate than the non-expert physician to classify the infarct location when both were compared to the experts' evaluation. Both the non-expert physician and the automated reports agreed with the experts' evaluation in most of cases (71 and 88%, respectively). The non-expert physician was "in partial agreement" with the experts in 39 cases (25%), and the automated generated reports, in 19 (12%). The non-expert physician was "in disagreement" with the experts in 6 cases

(4%), while the automated generated reports had no substantial disagreement with the experts' reports. The mean time of the non-expert physician evaluation was 1 min per scan, with the maximum time of 2.6 min.

**Prediction interpretability**. Instead of building black-box ML models, we aimed to provide interpretable models to elucidate whether the machine uses features of biological relevance. Figure 4 indicates the importance of features in the RF models. The most important feature was the percentage of injury of the region in question, followed by the injury of neighboring regions. This aligns with the correlations found between regional classification of injury by visual analysis (Table 4.I.a and II.a) and indicated that, in general, RF models and humans are using very similar features for scoring. The permutation feature importance test demonstrated the consistency of the importance of features learned in the training set, and their generalization in the testing set.

While these methods expose general features implied in the classification, it is important to highlight the reasoning of the prediction at individual level. This serves as validation for the ML models, as well to facilitate the calculation of treatment-relevant scores (e.g., ASPECTS) that depend on the reliable identification of injured regions. Therefore, SHAP was implemented in ADS to explain how the features were considered by the pre-trained model to predict infarct location in any given new sample. The Supplementary Fig. 3 illustrates one example of the explanation of our pre-trained model, which is outputted together with the regional predicts of the infarct location and their probabilities (Fig. 3 and Supplementary Note 1).

## Discussion

We created a fully automated system to quantify ischemic infarcts and report their location, with accuracy comparable to an expert evaluator, and among the inter-evaluators variation. The system is robust to major technical, lesion, and populational variations, and in an external unrelated population. The random forest (RF) models achieved the best performance in virtually all the regions (Table 3). The RF performance was followed by that of the binary threshold method, BT. However, although the BT accuracy was particularly high in areas with severe class imbalance (e.g., ACA), the general BT performance and its precision, in particular, were significantly lower than that of RF. This indirectly points to the ways AI uses the image features (in this case, the QFVs) to make a prediction. As confirmed by the feature analysis, the main feature determining the injury of a region is, as expected, the proportion of the respective region affected by the infarct. However, joined injury of neighboring regions have a secondary but still important effect in the decision (Fig. 4). Similarly, the human prediction also relays on these joined conditions in which the determination of injury in a given ROI is mostly correlated to its respective and dominant QFV component, followed by the components of the neighboring ROIs (as depicted in Table 4.II.c). Therefore, it is expected that more complex models, that take in account the

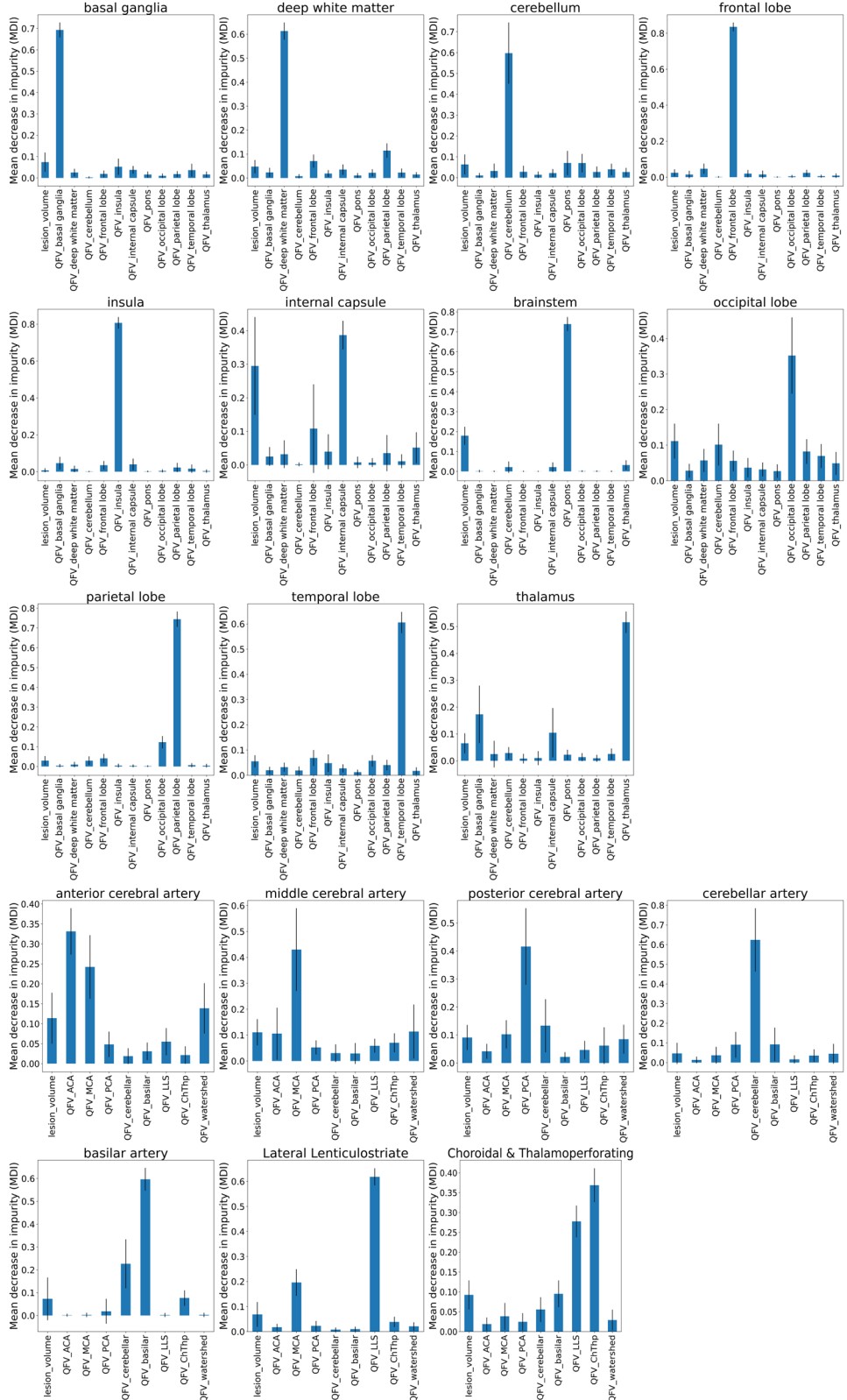

**Fig. 4 Feature importance, as revealed by the Mean Decrease in Impurity (MDI) of the Random Forest (RF) models (*n* = 1414).** The MDI is proportional to the importance of the features (the QFVs and lesion volume, in the *x*-axis) to predict the injury of the region in question (title of each graph). The QFVs represent the proportion of each ROI affected by the infarct. Note that the dominant QFV component agrees with the prediction of injury in the corresponding region and is followed by the QFV of its spatially neighboring regions.

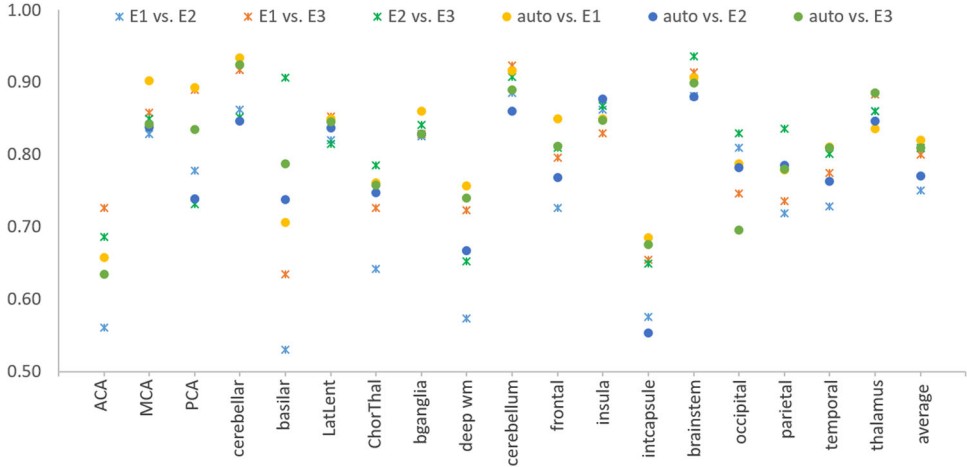

**Fig. 5 Agreement between human and machine on the definition of infarct location in the testing set (n = 464).** Intraclass correlations (ICCs, y-axis) among human evaluators (E1, E2, E3) and among evaluators and our automated model for infarct location classification (auto). ICC = 1 is perfect agreement.

joined probabilities, will generally have better performance than the simple BT for single ROIs.

From the biological point of view, this phenomenon likely relates to the coalescence of ischemic strokes, that do not respect the semantically-defined boundaries, particularly those defined by the classical structural atlas. From the technical point of view, inaccuracies in brain mapping can lead to mismatches between ROIs and the structures they define. This impacts the quantification, specifically when the injury is located in small ROIs, close to the ROI boundaries, or in mesial and periventricular areas. These areas are particularly challenging for the linear mapping in populations with high frequency of hydrocephalus or midline shifts, as occurs in acute stroke. For example, Fig. 3 shows a visible mismatch between the atlas definition of the ventricles and mesial structures and the brain in question, due to ventricular compression and midline shift, caused by the infarct. Although this inaccuracy did not lead to disagreements between the automated and the human radiological report in this large infarct, it might be the case in more focal lesions. As noted, the complex models employed for the prediction help to diminish this problem.

Another practical strategy we implemented in our automated pipeline (ADS[10]) is the option to recalculate the QFVs using a non-linear mapping. This theoretically improves the match between the brain in question and the template, which would consequently result in more accurate classification of the stroke location. In fact, we observed slight improvement in the location classification of infarcts when using non-linear brain mapping (as shown in Supplementary Table 3). The mildness on improving might be attributed to the presence of previous strokes or microvascular diseases that often occur in this population. These abnormalities alter the brain anatomy and contrast, reducing the accuracy of the non-linear algorithms. On the other hand, the low degree of deformation elasticity of the non-linear algorithm employed and the low granularity of our ROIs likely prevented mismatches in the classification of the lesion location. Given the cost / benefit (the non-linear deformation takes about 3 extra minutes of processing time) the linear mapping is the default option in ADS[10], and the non-linear mapping is offered as an optional.

Regarding the regional accuracy of the prediction, the lowest BACC of all models was in ACA (BACC = 0.712), for the arterial territory scheme, and internal capsule (BACC = 0.752), for the classical structural scheme. Infarcts in these regions were less frequent in our sample (in agreement with the epidemiology of

ischemic strokes) which is a limitation for model training and testing. This will be ameliorated by increasing the dataset. In addition, these regions offer extra challenges for both humans and machine, either by having ambiguous / highly variable territories, like the ACA (Arterial atlas—NITRC. https://www.nitrc.org/docman/?group_id=1498), or by their ill-defined limits in low resolution clinical images or small volume, like the internal capsule.

The feature analysis enriched the AI models, increasing their interpretability and their potential usefulness. Therefore, our ADS[10] system is suited to output not only the radiological report in semantic format but also the list with the proportion of injury in each area defined (the QFVs), the regional prediction of injury and the prediction probability, as well as explanatory reports showing how the QFVs were combined to predict injury in each area (example of ADS outputs in Supplementary Fig. 3 and Note 1). The QFVs are computable data objects that might serve as lesion loadings for anatomico-functional studies or to train artificial intelligence models. In clinics, these interpretable reports may, theoretically, improve the reliability of the system[26–28] by increasing transparency, promoting trust, and indirectly serving as quality control. For example, clinicians may identify cases where the model overemphasizes or fails to consider important information. Further quantitative studies (e.g., measuring number of reports generated, turnaround times, and error rates), user surveys, comparison of clinical outcomes, and cost analysis will be needed to test the real impact in practical settings.

A final consideration regards to the producibility of the automated generated reports. Our tool is linked to a public and expandable dataset of clinical images[11], and therefore will be supported by a dynamic dataset whose radiological evaluation can be modified / refined over time. The tool is modular, therefore flexible and adaptable to changes, for instance, in brain mapping procedures or parcellation schemes (e.g., different ROIs can be easily adopted, either to test their clinical significance or to train models in order to provide different types of reports). It is easily linkable to other software for image analysis, for example those that work directly in MRI scanner outputs, making them compatible with our image inputs. Therefore, it can be theoretically installed in radiological reading settings. The inclusion of a module that accepts users' feedback for models' retraining, maintenance and quality control is in our future plan. It is also in our short-term plan leveraging cloud-based infrastructure which could allow easy and large-scale processing of clinical research data.

In summary, using the original DWI as input, we created a fully automated system that includes automatic detection and segmentation of ischemic injuries and outputs radiological reports, in addition to the previously reported[12] 3D digital infarct mask, infarct volume, and the feature vector of regions affected by the acute stroke. We speculate that the automated radiological reports might be superior to "non-expert" reports, based on the proof-of-concept comparison with the non-expert physician evaluation. This would be particularly relevant, and potentially time-saver, in centers that lack second readers or neuroradiologists full time in emergency service. However, large scale prospective tests are imperative both to prove the clinical impact as well as to optimize the technology presented here. So far, we limit our contribution on generating a publicly available system that produces computable data objects, runs in real time, in local CPUs of regular personal computers, with minimal computational requirements[10,12], and is accessible to non-expert users, fulfilling the conditions to perform large scale, reliable and reproducible clinical and translational research.

## Data availability

The data used in this study are available at https://www.icpsr.umich.edu/web/ICPSR/studies/38464[11]. these data is public and free and can be downloaded directly from this repository after signing the Disclosure of User Agreement. Note that these data include all the images, in native space and post- processed (mapped to common space), the annotation of the stroke core (in the DWI), and the demographic and clinical information. This enables easy validation and replicability test of the results presented here. The STIR data were used under approval from the STIR steering committee for the current study, and so are not publicly available. These data are however available from the STIR / Vista Investigators upon reasonable request to Dr. Marie Luby (lubym@ninds.nih.gov). The source data for this manuscript is in Supplementary Data 6.

## Code availability

The tool described in this study is publicly available at https://www.nitrc.org/projects/ads[10]. The source code is available at https://doi.org/10.5281/zenodo.556523[029].

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

## Acknowledgements

This research was supported in part by the National Institute of Deaf and Communication Disorders, NIDCD, through R01 DC05375, R01 DC015466, P50 DC014664 (A.H., A.V.F.), the National Institute of Biomedical Imaging and Bioengineering, NIBIB, through P41 EB031771 (M.I.M., A.V.F.), and the Department of Neurology, University of Texas at Austin, the National Institute of Neurological Disorders and Stroke, NINDS, National Institutes of Health, NIH (STIR / Vista Imaging Investigators).

## Author contribution

A.V.F. and C.-F.L. conceived and designed the study, analyzed, and interpreted the data, drafted the work. Y.Z., V.Y., R.L., V.F. analyzed the data. A.E.H. acquired part of the data and significantly revised the manuscript. M.I.M. revised the manuscript. The STIR and VISTA investigators provided part of the data.

## Competing interests

The authors declare the following competing interests: Michael I. Miller owns "AnatomyWorks". This arrangement is managed by Johns Hopkins University in accordance with its conflict-of-interest policies. The remaining authors declare no competing interests.

## Additional information

## on behalf of the STIR and VISTA Imaging investigators

Max Wintermark[9,10], Steven J. Warach[11], Gregory W. Albers[9], Stephen M. Davis[12], James C. Grotta[13], Werner Hacke[14], Dong-Wha Kang[15], Chelsea Kidwell[16], Walter J. Koroshetz[17], Kennedy Lees[18], Michael H. Lev[19], David S. Liebeskind[20], A. Gregory Sorensen[21], Vincent N. Thijs[22], Götz Thomalla[23], Joanna M. Wardlaw[24] & Marie Luby[17]

[9]Radiology, Neuroimaging and Neurointervention, Stanford University, Stanford, CA, USA. [10]Centre Hospitalier Universitaire Vaudois, Lausanne, Switzerland. [11]UT Southwestern Clinical Research Institute of Austin, Department of Neurology and Neurotherapeutics, UT Southwestern Medical Center, Austin, USA. [12]Departments of Medicine and Neurology, Melbourne Brain Centre at the Royal Melbourne Hospital, University of Melbourne, Melbourne, VIC, Australia. [13]Department of Neurology, University of Texas Health Science Center, Houston, TX, USA. [14]Department of Neurology, University of Heidelberg, Heidelberg, Germany. [15]Department of Neurology, Asian Medical Center, University of Ulsan College of Medicine, Ulsan, Korea. [16]Department of Neurology and the Stroke Center, Georgetown University, Washington, DC, USA. [17]National Institute of Neurological Disorders and Stroke (NINDS), National Institutes of Health (NIH), Bethesda, MD, USA. [18]Institute of Cardiovascular and Medical Sciences, University of Glasgow, West- ern Infirmary, Glasgow, UK. [19]Massachusetts General Hospital and Harvard Medical School, Boston, MA, USA. [20]UCLA Stroke Center, Los Angeles, CA, USA. [21]Siemens Corporate Research, Inc, Princeton, NJ, USA. [22]Laboratory of Neurobiology, Vesalius Research Center, VIB, Experimental Neurology and Leuven Research Institute for Neuroscience and Disease, Department of Neurology, University Hos- pital Leuven, Leuven, Belgium. [23]University Medical Center Hamburg-Eppendorf, Hamburg, Germany. [24]Brain Research Imaging Centre, Division of Neuroimaging Sciences, Centre for Clinical Brain Sciences, University of Edinburgh, Edinburgh, UK.

