## [Peer Review File · Communications Medicine]

Reviewers' comments:

Reviewer #1 (Remarks to the Author):

The aim of this paper is to propose automated radiologic reports for clinical acute stroke MRI. This is a quite interesting and important topic. However, there are several concerns on the reproducibility of this study, including no external validation, no multiple reader comparisons, and no clinical effects.

1. The dataset was collected consecutively or not? Describe in detail especially on the test set.
2. I suspect that authors need more metrics including ICC among more than three readers and ML models.
3. Cross-validation with repetition could be more sensitive to external dataset, which should be tested.
4. There was a lack of training details on hyperparameter optimization, training strategies, etc.
5. There is a lack of new technology which consisted with existing technologies in viewpoint of engineering. Therefore, this system should be tested prospectively for evaluating effects in actual clinical settings.

Reviewer #2 (Remarks to the Author):

In this work, the author proposed an end-to-end automated system that outputs radiological reports about ischemic strokes. The manuscript is well-written; the proposed components in the system are interesting (e.g., the interpretation module). While the proposed system could be a potential solution to resolve the issue of how to provide meaningful information to clinicians to treat patients with acute ischemic stroke, there are some unclear details in the manuscript. Here are the questions/suggestions:

1. Abstract: The authors claimed that the system requires minimal computational requirements. But there is missing information in the manuscript about the system configuration requirement and the analysis of resource usage using the proposed system (e.g., how much memory is needed, do I need a GPU, etc.)?
2. P.3 Section 2.1. "We included 1,878 MRIs with evidence ..." Can the authors clarify whether the MRIs are completely mutually exclusive? (There could be reoccurring stroke from the same patients)
3. P.4 Section 2.1. "The DWIs were mapped to a common..." Any additional voxel preprocessing steps are performed (e.g., voxel standardization)
4. P.5 Section 2.3. "The quantitative feature (QFV) extracted reflect ..." It is an interesting feature generation approach, which relies on the proportion of ROI voxels in the stroke area and the non-stroke area. There are few questions:
 - a. How good is the segmentation/identification of each region in the brain? That is, can authors provide more details about whether all 1,878 align well to the template in MNI space. This could be an important question to address because identifying accurately "different regions of the brain" is an important factor to develop realistic and accurate QFV value.

- b. There is only one single feature (i.e., QFV) associated to each region? Please clarify. Have authors considered using more advanced feature engineering techniques, such as deep learning, to generate more features to improve the model classification performance?
- c. Figure 3. Is the equation of QFV is “ $QFV = \text{“infarct volume/ROI volume”}$ ”. It seems this is not the case according to the table values in the Figure 3. Please clarify.
5. P.5 Section 2.3. “the number of voxels with ADC intensity $>0.0014\text{mm}^2/\text{s}$...” How did the authors come up with these two thresholds from CSF and non-CSF voxels?
6. P.5 Section 2.3. “Low XXXX indicates ventricular enlargement.” How low?
7. P.6 Section 2.4 & Figure 1. “All ML models were 5-fold cross ...” Are there any patients with cerebral white matter lesions or recurring strokes? This may affect the system classification performance. How did the authors address this?
8. P.6 Section 2.4 and Figure 1. “... via 10 repeat 5-fold cross validation.” Is 10 repeated cross-validations or 100 repeated cross-validations are used? (Figure 1 shows that 100 repeated cross-validations are used)
9. P.10 Discussion “The feature analysis enriched the AI models, ...” It is interesting that the authors are trying to solve the issue of black-box machine learning models by providing model interpretation output in the report. While this is an important component to enhance the usability of the systems, there are a few open questions that may be better addressed in order to really show that the proposed automated system are useful to clinicians:
- a. Have authors evaluated the usability of this system to the clinicians? Any usability analyses (e.g., surveys) are conducted to determine whether the proposed system, especially the model interpretation components, are useful to clinicians and can increase their work efficiency?
- b. The proposed interpretation component in the system utilizes the SHAP values and the “impurity-based” feature importance analysis. How usable are these technical driven values to the clinicians? For example, if the system generates Figure 4 (and supplementary Figures 2 & 3), will clinicians know how to interpret them and use the information from the figures to drive clinical decisions (e.g., treatment to stroke patients)?

Reviewer #3 (Remarks to the Author):

In this paper, an acute infarction reporting system is constructed based on a large dataset. The system can reach the level of human observers, and the system is made available to the public, which can promote related research. One question is how accurate the linear deformation of the DWIs is, and its implications for follow-up studies.

We thank the reviewers for their interest in our manuscript and their thoughtful comments that have allowed us to improve the paper. Our responses to the reviewers' comments are shown below (the reviewers' original comments are pasted in bold). The changes in the revised manuscript are tracked and commented with the reviewer number and comment number (e.g., R1.2 is comment 2 of reviewer 1).

Reviewer #1:

The aim of this paper is to propose automated radiologic reports for clinical acute stroke MRI. This is quite interesting and important topic. However, there are several concerns on the reproducibility of this study including no external validation, no multiple reader comparisons, and no clinical effects.

We thank the reviewer for identifying these issues. We believe they are all solved as detailed below and reflected in our manuscript revisions. Briefly:

- a) the external validation includes an independent testing set (not seen in the training or validation phases) of 464 cases. Note that these, as well as the training set (n=1414), are clinical cases admitted at hospitals emergency, and not filtered data collected for research, or even for trials. This increases the potential generalization of the developed tools. In addition, inspired by this reviewer's comments, we now test the models in an external and unrelated population, the STIR dataset.
- b) We now include and analyze new readings, from two other expert physicians.
- c) A proof-of-concept test of the direct impact of this technology in clinical settings is now shown.
- d) We note that everything used / derived from this study is public, and available for anyone to test. This includes the data (even the raw clinical images and annotations), the models generated, as well as the software for the whole analysis and generation of automated reports. These is all referred in the manuscript and allows anyone to replicate our results using our data or other datasets, which is the ultimate test of reproducibility.

1. The dataset was collected consecutively or not? Describe in detail especially on the test set.

We re-wrote and copied below the description of the dataset. As you can see, the dataset used is a collection of very mixed cases at the hospital front door; real clinical images and metadata, over 10 years, of patients admitted in multiple hospitals that are part of a National Stroke Center. This is the first large annotated dataset of brain MRIs ever publicly available, accessible at <http://dx.doi.org/10.3886/ICPSR38464>. Parenthetically, as of today, this dataset is one of the 20 finalists of the FASEB-NIH DataWorks competition (<https://www.heriox.com/dataworks>). The detailed description of the creation of this dataset, as well as post-processing procedures and metadata are a paper per se, currently under review in Nature Scientific Data. I included the pre-print for your convenience (also publicly available at <https://www.researchsquare.com/article/rs-1705779/v1>).

The testing set was randomly defined as mentioned in the manuscript. The most detailed description and quantification of the training and the testing subsets is in Table 1. We reinforce that the testing set is completely independent; it was not seen in the training or validation phases. In addition, we now add a second testing set, an external unrelated population (STIR), which is further detailed in response to your comment 3.

“This study included MRIs of patients admitted to the Comprehensive Stroke Center at Johns Hopkins Hospital with the clinical diagnosis of ischemic stroke, between 2009 and 2019. This is a subset of the "Annotated Clinical MRIs and Linked Metadata of Patients with Acute Stroke", a dataset that we publicly shared¹⁰. Briefly, the entire dataset consists of 2,888 multimodal clinical MRIs performed at the admission of patients with acute brain strokes, retrospectively archived over 10 years, organized under FAIR principles¹⁶. Of note, only patients with MRI diagnosis of acute stroke were included, which represents a subset of all "front door" hospital stroke patients. The dataset includes lesion segmentation, expert radiological description, patient demographic information, and basic clinical profile. Details of this publicly available dataset are in the documentation that accompanies the data¹⁰ and in the related publication.¹⁷

In this study, we included 1,878 MRIs with evidence of ischemic stroke in the diffusion weighted images (DWI). The flowchart for data inclusion is shown in Figure 1. The data were random split into training set (n=1414, 75%) and testing set (n=464, 25%). The detailed description of the demographic, lesion and scanner profiles of the data used in this study is in Table 1. The distribution of infarcts according to brain location and the demographic characteristics reflect the general population of stroke patients. MRIs were obtained on eleven scanners from four different manufacturers, in different magnetic fields (1.5T and 3T), with dozens of different protocols. The DWIs had high in plane (axial) resolution (1.2x1.2mm, or less), and typical clinical high slice thickness (up to 5 mm plus gap). Although a challenge for imaging processing, the technical heterogeneity promotes the potential generalization of the resulting developed tools.”

“Our testing set was completely independent and unseen in the machine learning training and validation phases. We reinforce that although we used data from a single National Stroke Center, these data originated from multiple hospitals and a large geographic region, representing an "unfiltered, front-door hospital" sample and reflecting the profile of the national population with stroke. Yet, a second external testing set (STIR), was used to test the generalization of our models in a completely unrelated population. We have complied with all relevant STIR regulations for data usage.”

2. I suspect that authors need more metrics including icc among more than three readers and ml models.

Thank you for this suggestion. We asked two other highly qualified experts in the field (in addition to the original expert readers) to perform their radiological description, independently. We present the indices of agreement among all the evaluators, as well as between the evaluators and the machine learning classifications. We found that the automated method performs within the inter-evaluators' performance; for a summary please see Figure 3 and Supplementary Table 10. This is all described in the revised manuscript in the new Methods section “Multiple Evaluators Descriptions”, and in the Results section “Accuracy of ML models to predict infarct location”, 3rd paragraph, as copied below.

“To access the level of variation in visual descriptions, and the agreement of different evaluators with the developed automated reports, two other clinical experts, a neuroradiologist (VY) and a neurologist (RL), with more than 10 years of experience in stroke care and image reading, classified the infarct location in the whole testing set (n=464), following the same procedures described above. The comparison among the three evaluators, and among the evaluators and the automated classification

was made by the intraclass correlation (ICC) using the function ICC3 of the Python package “pingouin.intraclass-corr”²⁰.

“The automated classification of infarct location was also robust when compared with that of different experts. The mean ICCs of the model against each of the three evaluators were 0.82 ± 0.08 , 0.77 ± 0.11 , and 0.81 ± 0.08 , which rivaled to the ICCs among pairs of inter-evaluators: 0.75 ± 0.12 , 0.8 ± 0.09 , and 0.81 ± 0.08 ; with standard deviations consistently lower. The indices of agreement are presented in details and categorized by location in Supplementary Table 10. The regional distribution of ICCs was consistent inter-evaluators and between the model and the evaluators, i.e., regions with the lowest concordance among the model and the evaluators (e.g., ACA and internal capsule) also had the lowest concordance inter-evaluators (Supplementary Table 10 and Figure 3). Of note, these regions correspond to those with lowest lesion frequency or those with unclear or less consensual boundaries.”

3. Cross validation with repetition could be more sensitive to external dataset, which should be tested.

Thank you for the suggestion. We would like to clarify that our testing set was completely independent and unseen in the machine learning training and validation phases. To provide transparent optimization details, we revised the section “Machine Learning (ML) classification models to predict infarct location” as follows:

“All ML models were cross validated over the training set (1414 subjects, 75%) for hyperparameter searching and tested in the completely independent testing set of 464 subjects (flowchart in Figure 1). We performed 5-fold cross validation on the training set, for a large set of searching parameters. The models' hyperparameters with the top 3 performances (according to the sum of balanced accuracy (BACC) and F1 score, from this first run of 5-fold cross validation) were further determined and selected via 10 repeat 5-fold cross-validation, on the training set. The details of searching parameters' sets, final optimal parameters, cross validation results, and the definitions of performance metrics are in the Supplementary Table 1.”

In addition, inspired by this reviewer’s comments and aligned with our aim of pursuing realistic prospect of generalizability and reproducibility, we include in this revision the performance of our model in an external unrelated population, STIR (<http://stir.dellmed.utexas.edu>). This test is reported in the Methods (copied below) and the results are summarized in Supplementary Table 9. As you can see, “The model was robust in the external population, STIR, demonstrating similar performance to that achieved in our independent testing set”

4. There was a lack of training details on hyperparameter optimization, training strategies, etc

As per our response to your comment 3 above, these details were added in the text and reported in the Supplementary Materials, Table 1.

5. There is a lack of new technology which consisted with existing technologies in view point of engineering. Therefore this system should be tested prospectively for evaluating effects in actual clinical settings.

We agree with the reviewer that no new technology, from the mathematical or computational point-of-view, was developed. Rather, the novelty consists in the way we ensembled and adapted the existent methods to create an accessible tool that performs radiological interpretation and provides quantitative reports of lesion loadings readily useful as computational data objects to researchers of diverse expertise, from clinical research to AI. The novelty is also in the distribution model, as a public, free, and user-friendly tool, accessible to anyone performing medical, basic, or computational research.

We believe the technology reported and made available here will impact, in the short and medium term, translational research. To test the impact in clinical scenarios is a long-term plan, which depends on technical and bureaucratic procedures beyond the scope of the present study, as technology clearance by competent boards. Our conclusion, in the last paragraph (copied below) states this view. On the other hand, we understand the reviewer's reasoning and agree that at least an idea of prospective clinical importance should be given to the readers. Therefore, for the sake of scientific interest, we investigated how our method performs compared to an untrained physician (emergency doctor, with no formal training in neuro-radiology or neurology). This test (copied below) is now included in the paper. Of note, we were careful in our conclusion to clearly state that the real impact in clinical scenarios is speculative at this point, and is an answer to be pursued after the publication of this method.

In Methods, section 2.3

"A second question is whether the automated radiological reports would aid the flow of clinical stroke care, particularly in settings that do not count on highly trained experts or second radiology readers full time in emergency service. Testing clinical impact is beyond the scope of this paper, as it depends on further stages of technical and bureaucratic technology development. Even so, as proof-of-concept, we asked one emergency room physician, not formally trained in neuroradiology or neurology (VF), to classify the stroke location in a testing subset (n=155), again using the same procedures described above. The results of the here called "non-expert physician" and the automated radiological reports were compared to the expert physicians' readings, and rated as "in agreement", "in partial agreement", or "in disagreement" with those. "In disagreement" was used if an infarcted area was not or was wrongly described, and that would have clinical implications, such as change of clinically relevant metrics (e.g. ASPECTS). "In partial agreement" was used if the error would have no potential clinical implications. We also recorded the time for the "non-expert physician" reading."

In Results

"Our system was more accurate than the non-expert physician to classify the infarct location when both were compared to the experts' evaluation. Both the non-expert physician and the automated reports agreed with the experts' evaluation in most cases (71% and 88%, respectively). The non-expert physician was "in partial agreement" with the experts in 39 cases (25%), and the automated generated reports, in 19 (12%). The non-expert physician was "in disagreement" with the experts in 6 cases (4%), while the automated generated reports had no substantial disagreement with the experts' reports. The mean time for the non-expert physician evaluation was 1 minute per scan, with the maximum time of 2.6 minutes."

Last Paragraph

“We speculate that the automated radiological reports might be superior to "non-expert" reports, based on the proof-of-concept comparison with the non-expert physician evaluation. This would be particularly relevant, and potentially time-saver, in centers that lack second readers or neuroradiologists full time in emergency service. However, large scale prospective tests are imperative both to prove the clinical impact as well as to optimize the technology presented here. So far, we limit our contribution on generating a publicly available system that produces "computable data objects", runs in real time, in local CPU or GPU computers, with minimal computational requirements, and is accessible to non-expert users, fulfilling the conditions to perform large scale, reliable and reproducible clinical and translational research.”

Reviewer #2 (Remarks to the Author):

In this work, the author proposed an end-to-end automated system that outputs radiological reports about ischemic strokes. The manuscript is well-written; the proposed components in the system is interesting (e.g., the interpretation module). While the proposed system could be a potential solution to resolve the issue of how to provide meaningful information to clinicians to treat patients with acute ischemic stroke, there are some unclear details in the manuscript. Here are the questions/suggestions:

1. Abstract: The authors claimed that the system requires minimal computational requirements. But there is missing information in the manuscript about the system configuration requirement and the analysis of resource usage using the proposed system (e.g., how much memory is needed, do I need a GPU, etc.)?

The detailed technical description of our tool for Acute strokes detection and segmentation, ADS, is in our previous publication, referred to in this manuscript, and in its documentation in NITRC. In the present study, we are reporting the addition to ADS of a function to generate the “data computable objects” and radiological reports. This function only requires a couple of extra python module installations, such as Shap, under the same hardware environment. A list of python dependencies (also free and publicly available) and the installation instructions are given with codes in NITRC.

In summary, the computational requirements are the same as those fully described in our previous paper, as well as in the documentation that accompanies the tool, public available at NITRC. We can add all this information to the Supplementary Material if the reviewer thinks it is crucial. However, we would prefer not to do so to avoid verbose and redundancy with the previous report and the documentation readily available to users at download. If the reviewer agrees, our strategy was to clarify in the manuscript that the system runs in CPU or GPU, within the memory of a regular PC, and refer to the detailed sources of documentation mentioned above every time we talk about computational requirements.

2. P.3 Section 2.1. “We included 1,878 MRIs with evidence ...” Can the authors clarify whether the MRIs are completely mutually exclusive? (There could be reoccurring stroke from the same patients)

The MRIs are completely mutually exclusive; there is no reoccurring stroke from the same patients. We added this information in the paper.

3. P.4 Section 2.1. “The DWIs were mapped to a common...” Any additional voxel preprocessing steps are performed (e.g., voxel standardization)

The mapping to a common space (MNI) was enough to produce standardized feature vectors. For the most detailed information about parameters, each step of this procedure, and quality control, we refer the readers to our publication describing the dataset (the pre-print, included here for your convenience, is currently under review in Nature Scientific Data). We included this information in the manuscript.

“The detailed description of these procedures, including used parameters and quality control of the image mapping, is in our publication describing the creation of the dataset¹⁷”.

4. P.5 Section 2.3. “The quantitative feature (QFV) extracted reflect ...” It is an interesting feature generation approach, which relies on the proportion of ROI voxels in the stroke area and the non-stroke area. There are few questions:

a. How good is the segmentation/identification of each region in the brain? That is, can authors provide more details about whether all 1,878 align well to the template in MNI space. This could be an important question to address because identifying accurately “different regions of the brain” is an important factor to develop realistic and accurate QFV value.

The reviewer raised a very important point as the quality of the image mapping clearly impacts the structural definition of the brain and, therefore, affects the automatic labeling. Although this is a comprehensive question, the investigation of segmentation quality is complicated by the lack of the ground truth to define the boundaries of the “regions of the brain”. Therefore, the farthest one can go is to show indices of agreements (as Dice) for structures with clear boundaries such as the global brain shape or the ventricles.

We included a section “QUALITY CONTROL INDEX (QCI)” in the Supplementary Material that describes our quantitative procedures (in addition to the qualitative procedures of visual analysis) for quality control of the image mapping. We also had done similar procedures before, in our previous paper (attached to this submission) that describes the dataset and the “post-processing” procedures. For your convenience, I copied below Figure 6 of the referred paper, that summarizes the results. As you can see, a small minority of voxels were outer or inner the template brain contour, when considering a stringent bandwidth of 5 voxels. Importantly, the small ratio of error was stable over magnetic field and scan manufacturer.

b. There is only one single feature (i.e., QFV) associated to each region? Please clarify. Have authors considered using more advanced feature engineering techniques, such as deep learning, to generate more features to improve the model classification performance?

The QFV (which includes the proportion of each ROI infarcted and the total infarct volume) was the only feature included as it represents the biologically relevant information for lesion localization.

We considered to use Deep Learning, but as we now add in the manuscript:

“Because some ROIs are injured only in a few cases (see Table 1), deep neural network models would suffer from imbalanced classes issue over our large dataset. In addition, the interpretation of deep learning models is not as straightforward as that of classic ML methods. Furthermore, given that the ML models employed proved sufficiently efficient compared to humans (as we will show in Results), we opted to further investigate the potential of deep learning techniques to improve the classification performance when this dataset is expanded, or when other public sets of clinical data become available.”

c. Figure 3. Is the equation of QFV is “QFV = “infarct volume/ROI volume”. It seems this is not the case according to the table values in the Figure 3. Please clarify.

We apologize: the values in Figure 3 were wrong. It seems that we used an old version of our calculation and oversaw this mistake when revising the paper. We corrected this mistake, and we confirm that the QFVs represent the proportion of each ROI injured. We also added to the text that “all ROIs are bilateral (except by the "brainstem") and homologous ROIs have approximately the same volume.”

5. P.5 Section 2.3. “the number of voxels with ADC intensity >0.0014mm/s2...” How did the authors come up with these two thresholds from CSF and non-CSF voxels?

The correct value is 0.0018 mm²/s (we corrected this information in the text), and it is based on previous literature, now referenced in the paper (reference 22). We thank the reviewer for pointing out this mistake and the lack of references.

6. P.5 Section 2.3. “Low XXXX indicates ventricular enlargement.” How low?

Thank you for pointing out the unclearness of this statement. We meant ventricles larger than the average population with visually judged “normal” ventricles. We did not mean abnormally enlarged, nor did we establish any threshold for abnormal enlargement. We clarify this sentence as follows: “... indicates ventricular enlargement compared to the expected average ventricle size (although not necessarily hydrocephalus)”

7. P.6 Section 2.4 & Figure 1. “All ML models were 5-fold cross ...” Are there any patients with cerebral white matter lesions or recurring strokes? This may affect the system classification performance. How did the authors address this?

Yes, because this is an elderly population with vascular disease, many patients had white matter microvascular lesions, and a few patients even had previous strokes. However, since we are using linear mapping, the possible contrast variations have no significant effect in the deformation quality (please, also see our response to your comment 4a, about mapping quality control). On the other hand, in our test using non-linear deformation (now included in the revised paper), we observed agreement indices for classification of lesion location similar to those of linear deformation, and no drastic errors in brain mapping were found in the quality control. We speculate (3rd paragraph of the Discussion, copied below) that the mild/moderate degree of elasticity of DiPy, summed to the low granularity of our labels, hindered errors. Finally, we note that previous lesions and drastic intensity abnormalities could affect the lesion segmentation. However, this study utilized manually segmented lesions, and focused on the location classification, not on lesion segmentation. Still, if users want to use our completely automated pipeline (that includes lesion segmentation), we had proven before that our algorithm for segmentation of the stroke core is successful even in the presence of associated vascular disease (Liu et al, Comm Medicine 61:1, 2021).

“Another practical strategy we implemented in our automated pipeline (ADS) is the option to recalculate the QFVs using a non-linear mapping. This theoretically improves the match between the brain in question and the template, which would consequently result in more accurate classification of the stroke location. In fact, we observed slight improvement in the location classification of infarcts when using non-linear brain mapping (as shown in Supplementary Table 8). The mildness on improving might be attributed to the presence of previous strokes or microvascular diseases that often occur in this population. These join abnormalities alter the brain anatomy and contrast, reducing the accuracy on the non-linear algorithms. On the other hand, the low degree of deformation elasticity and the low granularity of our ROIs likely prevented mismatches in the classification of the lesion location. Given the cost/benefit (the non-linear deformation takes about 3 extra minutes of processing time) the linear mapping is the default option in ADS, and the non-linear mapping is offered as an optional.”

8. P.6 Section 2.4 and Figure 1. “... via 10 repeat 5-fold cross validation.” Is 10 repeated cross-validations or 100 repeated cross-validations are used? (Figure 1 shows that 100 repeated cross-validations are used)

We apologize and thank the reviewer for pointing out this typo. It is 10 repeats 5-cross validation. We modified Figure 1 accordingly.

9. P.10 Discussion “The feature analysis enriched the AI models, ...” It is interesting that the authors are trying to solve the issue of black-box machine learning models by providing model interpretation output in the report. While this is an important component to enhance the usability of the systems, there are a few open questions that may be better addressed in order to really show that the proposed automated system are useful to clinicians:

a. Have authors evaluated the usability of this system to the clinicians? Any usability analyses (e.g., surveys) are conducted to determine whether the proposed system, especially the model interpretation components, are useful to clinicians and can increase their work efficiency?

This is a very relevant question, also asked by reviewer 1. We would like to respectfully refer this reviewer to our response to comment 5 of reviewer 1.

b. The proposed interpretation component in the system utilizes the SHAP values and the “impurity-based” feature importance analysis. How usable are these technical driven values to the clinicians? For example, if the system generates Figure 4 (and supplementary Figures 2 &3), will clinicians know how to interpret them and use the information from the figures to drive clinical decisions (e.g., treatment to stroke patients)?

Our system outputs supplementary figures 3 and 4. The top text in Supplementary figure 4 is the radiological report in the same format that clinicians receive from radiologists and use to make decisions in their daily practice. The quantitative tables in this figure report infarct volumes, one of the most important criteria for acute treatment. Supplementary figure 3 (created with “SHAP” results) is useful for validation, i.e., to show that the system is utilizing biologically relevant features for classification. This is important because, as you know, in the recent past several deep learning models had utilized non-biological “noise” to create (still efficient) classification models, which generated a certain degree of resistance to AI applications in the biomedical community. In addition to this validation, some urgent clinical decisions regarding stroke treatment or prognostic prediction are supported by clinical scores derived from the regional degree of injury (e.g. ASPECTS), which is outputted. This is now mentioned in the section “Prediction Interpretability”:

“While these methods expose the features implied in the classification at group level, it is important to highlight the reasoning of the prediction at individual level. This is important as a validation for the ML models, as well to facilitate the calculation of treatment-relevant scores (e.g., ASPECTS) that depend on the reliable identification of injured regions”

Reviewer #3 (Remarks to the Author):

In this paper, an acute infarction reporting system is constructed based on a large dataset. The system can reach the level of human observers, and the system is made available to the public, which can promote related research. One question is how accurate the linear deformation of the DWIs is, and its implications for follow-up studies.

We thank the reviewer for this relevant comment, which was also made by reviewer 2. We respectfully refer this reviewer to our response to comment 4a of reviewer 2, in which we discuss the accuracy of linear transformation. Furthermore, inspired by your comment, we tested whether a non-linear transformation (with Dipy) would increase the accuracy of our models. The results are now shown in the new Supplementary Figure 8. As expected, there is a trade-off between accuracy and time for image processing. Although we keep the linear transformation as the default option of our pipeline, we now offer users the option of performing non-linear mapping, which is particularly useful if the quality control index of linear mapping is beyond the expectation for a specific subject / study. We added this information in our Methods and Discussion sections, as follows:

“To access the accuracy of the QFVs extracted, we deliver quality control indices (described in the supplementary material) that indicate the agreement between the contour of the brain in question and the atlases in which the brain structures are defined. Lastly, we extracted QFVs from brains non-linearly mapped to the atlases (with Dipy), to evaluate the influence of the brain mapping method (linear vs. non-linear) in the automated prediction of stroke location.”

“Another practical strategy we implemented in our automated pipeline (ADS) is the option to recalculate the QFVs using a non-linear mapping. This theoretically improves the match between the brain in question and the template, which would consequently result in more accurate classification of the stroke location. In fact, we observed slight improvement in the location classification of infarcts when using non-linear brain mapping (as shown in Supplementary Table 8). The mildness on improving might be attributed to the presence of previous strokes or microvascular diseases that often occur in this population. These join abnormalities alter the brain anatomy and contrast, reducing the accuracy on the non-linear algorithms. On the other hand, the low degree of deformation elasticity and the low granularity of our ROIs likely prevented mismatches in the classification of the lesion location. Given the cost/benefit (the non-linear deformation takes about 3 extra minutes of processing time) the linear mapping is the default option in ADS, and the non-linear mapping is offered as an optional.”

REVIEWERS' COMMENTS:

Reviewer #1 (Remarks to the Author):

The paper has been improved significantly. However, there are still issues on the producibility and lack of new methodology.

Reviewer #2 (Remarks to the Author):

The authors have addressed all my questions. Thank you very much.

One minor note is that the usability of this "interpretable" system remains an interesting research question (only one non-export physician was used in the evaluation). It would be very interesting to understand how interpretable AI can increase the acceptability of AI in physician practice (which is out of the scope of this paper. Still, it is a possible and valuable future work).